



**Decoding multicomponent hydrochemical anomalies: A synergy**
**anomaly detection model for earthquake forecasting in active**
**tectonic zone**
Weiye Shao[1], Ying Li[2,*], Xiaocheng Zhou[2,*], Zhi Chen[2], Huajiao Liu[3], Zhaofei Liu[2], Chang Lu[2],
Yuwen Wang[2,4], Zhaojun Zeng[2,4], Yun Wang[5], Hongyi He[6,2], Shaohui Fan[1]
[1]Yunnan Earthquake Agency, Kunming 650224, China
[2]United Laboratory of High-Pressure Physics and Earthquake Science, Institute of Earthquake Forecasting, Beijing
100036, China
[3]Sichuan Earthquake Agency, Chengdu 610041, China
[4]School of the Earth Sciences and Resources, China University of Geosciences, Beijing 100083, China
[5]Department of Insurance, School of Finance, Yunnan University of Finance and Economics, Kunming 650221,
China
[6]Key Laboratory of Shale Gas Exploration, Ministry of Natural Resources, China University of Petroleum, Beijing
102249, China
*Correspondence to*: Ying Li (liying@ief.ac.cn), Xiaocheng Zhou (zhouxiaocheng188@163.com).





**Abstract**

18        The intersection of the Xiaojiang Fault and the Red River Fault at the southeastern margin of the

Tibetan Plateau experiences intense tectonic activity. At this intersection, frequent destructive
earthquakes have induced hydrochemical variations in thermal springs. In this study, Bayesian change
point analysis is applied, and a multicomponent synergy anomaly detection model is developed using
five years of monitoring data (2019–2024) from two thermal springs in the region to achieve real-time
forecasting of occurrence timing for $M \geq 4$ earthquakes. Comprehensive analysis demonstrates that the
anomaly detection model possesses reliable real-time anomaly detection capabilities. Tailored model
parameters for specific hydrochemical components account for their differences in response
characteristics to seismic activity. The model identifies $Na^+$, $Ca^{2+}$, $Cl^-$, $SO_4^{2-}$, $\delta D$, and $\delta^{18}O$ as sensitive
indicators for strong earthquake forecasting. The multicomponent synergy alarm mechanism for
hydrochemistry overcomes the limitations of single-parameter methods, which significantly enhances
the model's overall performance in earthquake forecasting. The number of hydrochemical components
with synchronous anomalies serves as a reliable criterion for determining alarm intensity, with higher
intensity typically correlating with larger earthquake magnitudes or shorter epicentral distances.
**Keywords:**
Thermal springs, Hydrogeochemistry, Multicomponent synergy, Anomaly detection model,
Earthquake forecasting



## 1. Introduction

Earthquake forecasting, a frontier in geosciences, relies on detecting sensitive and reliable precursor anomalies (Chen, 2009; Pritchard et al., 2020). Subsurface fluids, owing to their ease of migration and incompressibility, respond rapidly to dynamic crustal stress changes during earthquake preparation. These responses often result in significant changes in the physicochemical properties of the fluids (Lee et al., 2017; Gori & Barberio, 2022; Tian et al., 2023). Moreover, subsurface fluids can transmit deep geological signals to the surface, for example, through thermal springs, which makes them valuable targets for monitoring precursor anomalies. Currently, earthquake-related anomalies in subsurface fluids are widely monitored across various spatial and temporal scales. These include hydrological anomalies such as water temperature, water level, and flow rate (Shi et al., 2015; Lee et al., 2017; Petitta et al., 2018; Di Matteo et al., 2020; Du et al., 2023), hydrogeochemical anomalies like major elements, trace elements, and stable isotopes (Ide et al., 2020; Nakagawa et al., 2020; Barbieri et al., 2021; Wang et al., 2021; Zhang et al., 2021; Yan et al., 2022), and gas geochemical anomalies such as radon, helium, and carbon dioxide (Chaudhuri et al., 2011; Fu et al., 2017; Woith et al., 2020; Zhao et al., 2021; Zhou et al., 2021). While some fluid precursor anomalies have shown predictive value, many are still identified only through retrospective analysis after earthquakes. Moreover, continuous fluid monitoring data often reflect integrated signals from multiple sources, including seismic activity, environmental variability, and human-induced influences (Martinelli, 2020). The isolation of true seismic precursor anomalies from such complex datasets remains a significant challenge in current earthquake forecasting research.

In analysing large-scale fluid monitoring data, traditional anomaly detection methods typically



rely on manually defined fixed thresholds to identify fluctuations. Techniques such as trend analysis
and standard deviation methods offer clear advantages in capturing prominent anomalies (Ingebritsen
& Manga, 2014; Yan et al., 2018). However, in practice, fluid monitoring data often integrate
superimposed signals from both tectonic and non-tectonic sources and exhibit complex nonlinear
dynamic behaviors. These characteristics present notable limitations for traditional statistical
approaches in effectively identifying fluid precursor anomalies (Yan et al., 2021). Machine learning-
based anomaly detection algorithms offer new perspectives for earthquake forecasting by uncovering
hidden precursor signals within large volumes of monitoring data (Li et al., 2022, 2023). In recent
years, algorithms such as artificial neural networks, long short-term memory networks, and random
forests have been widely applied to anomaly detection in individual indicators, such as water levels
and radon concentrations, significantly enhancing detection accuracy and sensitivity (Tareen et al.,
2019; Haider et al., 2021; Feng et al., 2022; Zhang et al., 2025). However, single-indicator
measurements are easily influenced by meteorological, tidal, and other environmental factors. While
regression models and similar techniques have been used to correct these interferences, challenges
remain in effectively distinguishing non-seismic anomalies (Woith, 2015; Soldati et al., 2020).
Moreover, single-indicator analysis does not leverage the synergistic relationships among multiple
indicators, which thus limits its ability to enhance the reliability of anomaly identification.

Thermal springs are natural discharge outlets of deep-circulating groundwater and they offer

distinct advantages for hydrogeochemical monitoring. The hydrochemical components (e.g., $Na^+$, $Cl^-$,
$SO_4^{2-}$) of thermal springs tend to exhibit high stability, rapid upward migration, and limited
susceptibility to environmental interference. These characteristics help minimise non-seismic noise



and allow for a more accurate reflection of hydrogeological changes during earthquake preparation
(Martinelli, 2020; Tian et al., 2024). Numerous studies have reported diverse geochemical behaviors
among hydrochemical components, which shows marked differences in their response magnitude,
quantity, patterns, and timing to tectonic stress variations throughout the earthquake preparation
process (Li et al., 2021; Yan et al., 2022; Tian et al., 2023). Therefore, applying anomaly detection
algorithms to evaluate the abnormal response characteristics of individual hydrochemical components
and integrating multiple components to enhance anomaly identification accuracy may represent a
promising technical approach for precursor recognition. Current research on hydrochemical anomaly
detection algorithms remains in an exploratory stage (Castellana & Biagi, 2008). Existing studies have
demonstrated the effectiveness of common machine learning algorithms in identifying abnormal
periods in hydrochemical data while also emphasising the need for scenario-specific optimisation of
key indicators (Zhu et al., 2024). However, there is an urgent need to investigate the synergistic
anomaly response patterns among hydrochemical components and to identify sensitive indicators for
reliable forecasting of strong earthquakes.

This study focuses on the tectonically active region at the intersection of the Xiaojiang Fault (XJF)

and the Red River Fault (RRF) on the southeastern margin of the Tibetan Plateau. The real-time
anomaly was innovatively modified using the detection algorithm developed by Piersanti et al. (2016),
and its application was extended to the analysis of multiple hydrochemical components in thermal
springs across the study area. By integrating continuous monitoring data of hydrochemical ions and
hydrogen–oxygen isotopes with earthquake catalogs and applying Bayesian change point (BCP)
analysis, this study optimised parameters for specific components and built a multi-component joint



anomaly detection model. This model supports anomaly detection in both long-term time series and
real-time earthquake forecasting across different time scales. The main objectives of this study are as
follows: (1) to evaluate the applicability and performance of the algorithm in analysing hydrochemical
time series; (2) to identify effective hydrochemical indicators for forecasting strong earthquakes in the
study area; and (3) to assess the feasibility of the multi-component joint anomaly detection model and
explore the relationship between hydrochemical variations and seismic activity by analysing the
number of components with synchronous anomalies.

**2. Geological setting**
The southeastern Tibetan Plateau has undergone sustained rotation and southeastward extrusion,
driven by the collision-induced uplift and deformation between the Indian and Eurasian plates. The
rotation and extrusion effects have resulted in the formation of an active tectonic region characterised
by large-scale strike-slip fault systems and the presence of intracontinental microplates (Tapponnier et
al., 1982; Yin & Harrison, 2000; Xu et al., 2011) (Figure 1). Among these structures, the XJF and RRF
serve as key strike-slip boundaries and play critical roles in the tectonic evolution and material
extrusion of the southeastern Tibetan Plateau (Zhang et al., 2003; Tong et al., 2015). The intersection
area of the XJF and RRF serves as the frontal zone accommodating the extrusion of the Sichuan–
Yunnan Block (SYB). The XJF is blocked by the Indochina Block (ICB) and has not yet propagated
southward through the RRF, which makes the intersection area the primary zone of stress accumulation,
where ongoing dextral compressional motion of the SYB occurs (Wen et al., 2022; Li et al., 2024;
Shao et al., 2024). The deeply incised XJF and RRF, along with secondary faults such as the Qujiang



Fault (QJF) and Shiping–Jianshui Fault (SJF) in this region, act as conduits for deep-circulating
thermal waters and the exchange of seismic information, with thermal springs commonly found along
these faults. This area experiences prolonged stress accumulation and intense tectonic deformation,
accompanied by historical moderate-to-strong seismic activity (Wen et al., 2008), which makes it a
critical zone for earthquake hazard monitoring. Consequently, this region is an ideal setting for
investigating how variations in hydrochemical compositions respond to seismic activity.

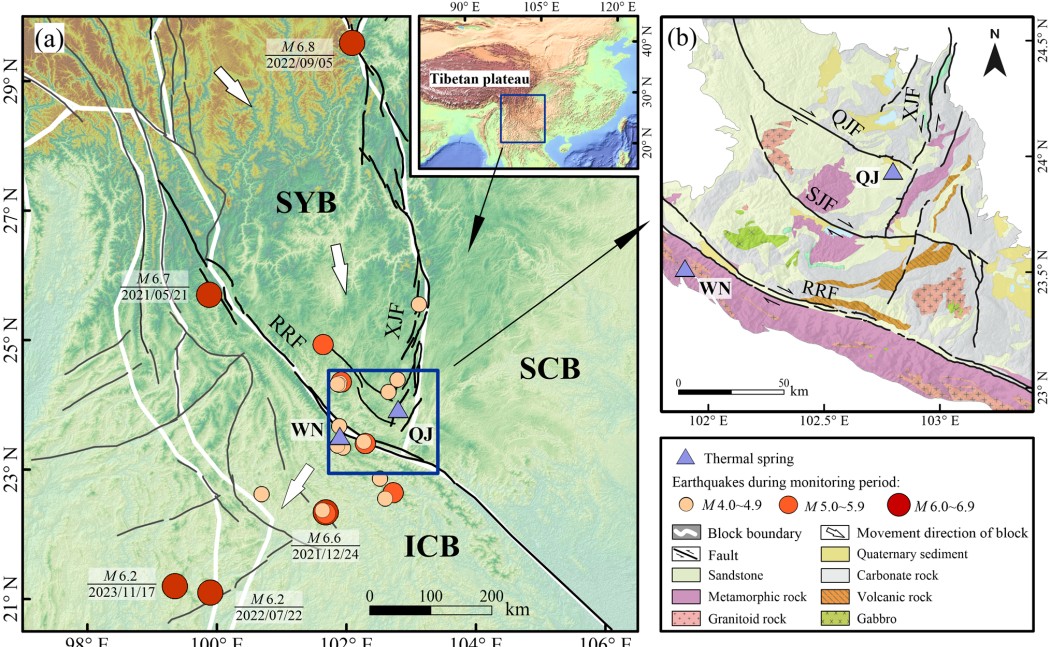


Figure 1. (a) Simplified tectonic map of the southeastern margin of the Tibetan Plateau, showing the distribution of
$M \geq 4$ earthquakes during the thermal spring monitoring period. (b) Locations of continuously monitored thermal
spring sites, fault distribution, and the geological map at the intersection of the Xiaojiang Fault (XJF) and Red River
Fault (RRF). Earthquake catalogues are obtained from the China National Earthquake Data Center
(https://data.earthquake.cn/). The tectonic divisions and active faults are sourced from Deng et al., 2002, and the
geological map is adapted from Ma et al., 2002. SYB: Sichuan–Yunnan Block; ICB: Indochina Block; SCB: South



China Block; QJF: Qujiang Fault; SJF: Shiping–Jianshui Fault.

This study involves continuous hydrochemical monitoring at two high-temperature springs,

Qujiang (QJ) and Wana (WN), located at key tectonic sites within the research area (Figure 1b). QJ is
situated at the intersection of the XJF and the QJF, with sandstone as the predominant country rock.
QJ is positioned at a critical location where the sinistral slip rate of the XJF decreases sharply from 8–
11 mm/a to approximately 4 mm/a after crossing the QJF (Wen et al., 2011; Wang et al., 2014). WN,
located along the RRF, is hosted by gneiss and mylonite and lies within a stress concentration zone,
where the SYB undergoes southwestward deflection, compressing the RRF (Schoenbohm et al., 2006;
Li et al., 2019; Wen et al., 2022). The two hot springs are located along the boundary faults that control
the regional tectonic pattern, and their hydrochemical variations may provide sensitive indicators of
changes in the earthquake preparation state within the intersection area.

**3. Data and methods**
**3.1. Thermal spring monitoring data**

The monitoring period for the QJ spring spanned from June 1, 2019, to May 21, 2024

(approximately 5 years), while the WN spring was monitored from October 3, 2021, to May 21, 2024
(approximately 2.5 years). Synchronous monitoring of hydrogen and oxygen isotopes was conducted
at both springs between January 1, 2023, and February 21, 2024. All monitoring parameters for the
thermal springs and their hydrochemical components were measured every three days. Water
temperature, pH, and electrical conductivity (EC) were measured *in situ* using a portable multi-
parameter water quality analyser (HQ40D, HACH, USA), with measurement accuracies of 0.1°C, 0.01,



and 1 μS/cm, respectively. Rainfall data were collected through continuous in situ monitoring using an
RTP-II tri-element meteorological instrument with a resolution of 0.1 mm. Before the thermal water
samples were collected, high-density polyethylene (HDPE) bottles were thoroughly rinsed three times
with deionised water and twice with thermal water. Water samples were then filtered through 0.45 μm
micropore membranes and stored in HDPE bottles. Samples intended for cation analysis were acidified
with high-purity nitric acid. During collection, care was taken to prevent the introduction of air bubbles,
and samples were immediately sealed hermetically for preservation.

The concentrations of major ions ($Na^+$, $K^+$, $Ca^{2+}$, $Mg^{2+}$, $Li^+$, $F^-$, $Cl^-$, $SO_4^{2-}$, $Br^-$, $NO_3^-$) were

analysed using a Thermo Scientific Dionex Aquion IC system equipped with an AS40 autosampler,
which had a detection limit of 0.01 mg/L. $HCO_3^-$ and $CO_3^{2-}$ concentrations were determined via
standard titration procedures using a ZDJ-3D potentiometric titrator with 0.05 mol/L HCl. $\delta^{18}O$ and
$\delta^2H$ values were determined using a Picarro L2140-i water isotope analyser, with precisions of 0.015 ‰
and 0.05 ‰, respectively. All analyses were conducted at the Key Laboratory of the Institute of
Earthquake Forecasting, China Earthquake Administration. The monitoring data are detailed in data
set S1. To ensure data accuracy, cation–anion balance error tests were performed for each sample, with
all ionic deviations kept within ± 5%. The ion balance error is calculated as below:
$$ib(\%) = \frac{\sum cations - \sum anions}{\sum cations + \sum anions} \times 100 \qquad (1)$$

**3.2. Earthquake data collection and processing**

The anomaly detection model developed in this study focused on forecasting destructive

earthquakes with magnitudes ($M$) ≥ 4. To identify earthquakes that might influence hydrochemical



component variations, while excluding those unrelated to precursors, and to establish a precise
correlation between changes in hydrochemical components and seismic activity, an earthquake
screening method based on the preparation zone radius formula (Dobrovolsky et al., 1979) was
employed:

$R = 10^{0.43M}$                                    (2)

where $M$ represents the earthquake magnitude, and $R$ denotes the radius (in km) of the earthquake
preparation zone.

Earthquakes were selected as study events based on the criterion that the epicentral distance ($\Delta$)

from the thermal spring monitoring sites did not exceed the earthquake preparation zone radius ($R$)
(Figure 1a). The QJ site was within the preparation zones of 22 $M \geq 4$ earthquakes during its monitoring
period (2019/06/01–2024/05/21), while the WN site was within the preparation zones of 12 $M \geq 4$
earthquakes during its observation period (2021/10/03–2024/05/21) (Table S1). All earthquakes had
focal depths ranging from 8 to 16 km, and they were classified as shallow-focus events. The earthquake
catalogue was obtained from the National Earthquake Data Center of China (http://data.earthquake.cn).

Seismic moment ($M_0$), which directly reflects fault geometry parameters and the rigidity of the

medium, accurately quantifies earthquake rupture processes and mechanical energy release. Compared
with magnitude scales, the seismic moment is more suitable for analysing the seismic impact on
hydrochemical component changes in thermal springs. The commonly used empirical formula for
estimating seismic moment based on magnitude (Hanks & Kanamori, 1979) is as follows:

$lgM_0 = 1.5M + 16.1$                                    (3)

Stress attenuates with increasing epicentral distance during the earthquake preparation process and





directly influences the development of thermal water seepage pathways and the intensity of water–
rock interactions (Wang & Manga, 2010; Ingebritsen & Manga, 2019). To account for distance-related
effects, the seismic moment requires correction using the following empirical formula (Piersanti et al.,

2016):

$$M_{0cor} = M_0 / \Delta^\omega \tag{4}$$
where $\omega$ is the weighting factor, in this study $\omega$ takes the value of 1.

**3.3. Hydrochemical component time series**
The geochemical behaviors of different components in thermal spring water show significant
variation, with each component exhibiting distinct characteristics in terms of anomaly amplitude,
temporal evolution, and precursor response sensitivity. Controlled by unique hydrogeological
conditions, the hydrochemical variations of each thermal spring also display spatial differences in
response to tectonic activity. To effectively extract anomalous signals, the anomaly responses of
different components and springs in the study area are compared, and the algorithm's generalisability
across springs is validated. This study establishes independent time series for each component at
different springs (Figure 2 and S1).



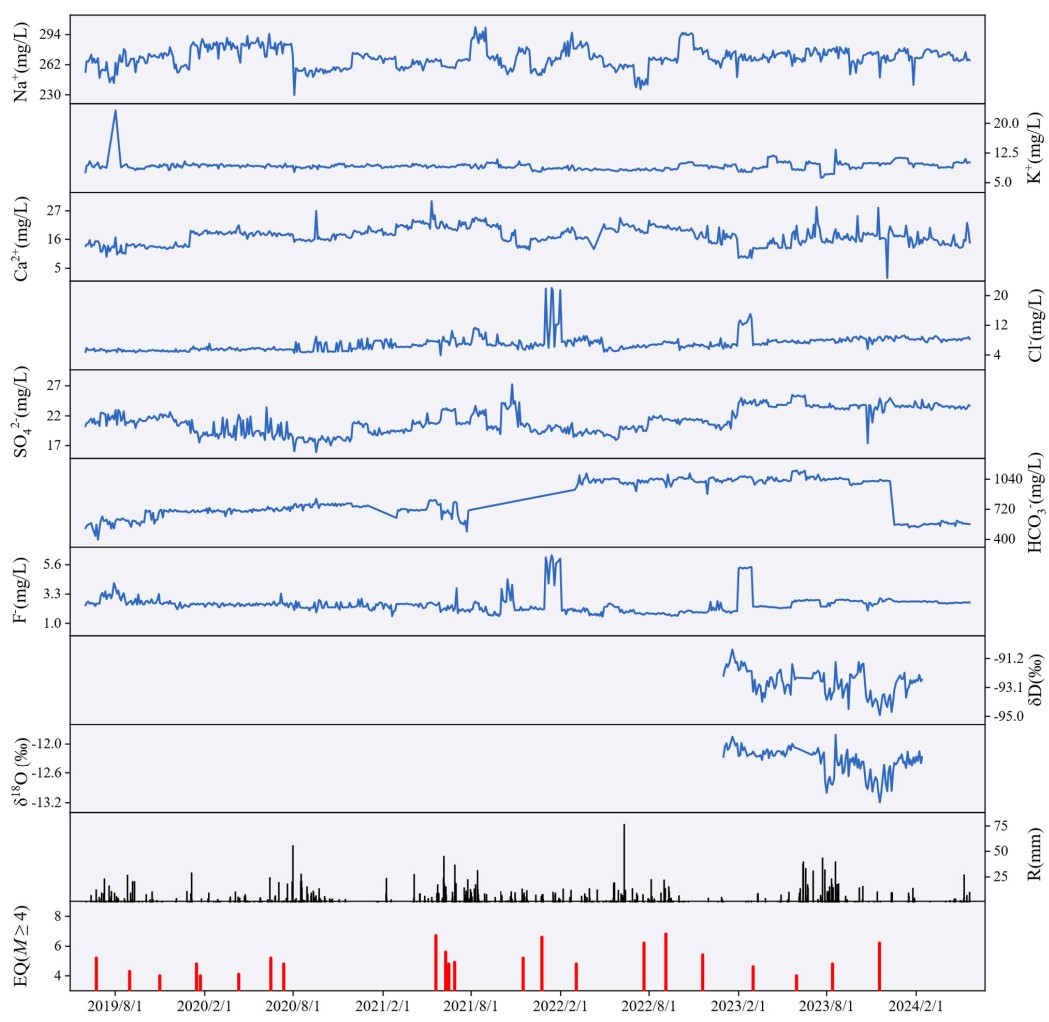


Figure 2. Time series of hydrochemical components at Qujiang spring, along with rainfall and earthquake events.

**3.3.1. Data evaluation and noise removal**
The dynamic variations in thermal spring ion concentrations are crucial for identifying seismic
precursors. When observed data show minimal fluctuations or remain consistently below detection
limits over extended periods, it becomes challenging to effectively extract hydrogeochemical anomaly
signals before an earthquake. Long-term monitoring reveals that concentrations of $Mg^{2+}$, $Br^-$, and
$NO_3^-$ are extremely low and remain consistently below instrumental detection limits, without temporal



fluctuations, which thus limits their value for tracking seismic precursors. Consequently, $Na^+$, $K^+$, $Ca^{2+}$,
$Cl^-$, $SO_4^{2-}$, $HCO_3^-$, $F^-$, $\delta D$, and $\delta^{18}O$ have been selected for earthquake anomaly identification owing
to their consistent continuity and reliable data characteristics.

The thermal spring water in the study area originates from atmospheric precipitation recharge. It

circulates deeply through faults, is heated by geothermal energy, and then discharges at the surface,
with its hydrochemical composition mainly determined by the lithology of the surrounding rocks (Shao
et al., 2024). Consequently, ambient temperature and atmospheric pressure at the spring outlet have a
negligible effect on the hydrochemistry. However, rainfall serves not only as the primary water source
but also accelerates groundwater circulation, promotes shallow infiltration, and mixes with thermal
waters (Taylor et al., 2012; Hosono et al., 2020; Colman et al., 2021). This process can potentially
obscure deep-seated earthquake preparatory signals carried by the thermal spring. Consequently, this
study focuses on assessing the potential perturbations induced by rainfall on thermal spring
hydrochemistry. Unlike temperature and pressure, rainfall causes pulsed disturbances, typically
manifesting as intermittent spikes followed by extended zero-value intervals in sampling data. To
suppress high-frequency noise from short-term environmental disturbances such as rainfall while
preserving mid- to low-frequency tectonic signals, a 15-day backward moving average is applied to
process the 3-day resolution hydrochemistry data:
$$MA(t) = \frac{1}{15} \sum_{t-14}^{t} Dr(t)$$
(5)

where $MA$ is the 15-day moving average, and $Dr$ is the daily raw data.
**3.3.2. Correlation analysis**

The influence of rainfall on the hydrochemical dynamics of thermal springs may exhibit a lag



effect, while hydrochemical precursor anomalies induced during earthquake preparation processes
typically precede earthquake events. These two mechanisms exhibit a significant temporal phase
difference in their perturbations to hydrochemical components. In this study, the cross-correlation
function is employed to quantitatively analyse the temporal offset between the impacts of rainfall and
seismic activity on thermal spring hydrochemistry. This study aims to identify the maximum
correlation time offset between rainfall-hydrochemistry and precursory anomaly-main shock events
via the calculation of correlation coefficients at varying lag times. The cross-correlation function is
defined as:
$$CC_{xy}(k) = \frac{1}{N} \sum_{t=1}^{N-k} (x_t - \overline{x})(y_{t+k} - \overline{y})$$
(6)

where $x$ and $y$ are two time series, $\overline{x}$ and $\overline{y}$ represent their sample means, $N$ is the series length, and
$k$ denotes the lag. Considering the seasonal effects of rainfall and the response time reliability of
seismic precursors, the $k$ is set within a range of −45 to 45 days.

In the cross-correlation analysis, the denoised hydrochemical component time series (processed

using a 15-day moving average) are correlated with both the rainfall time series and the distance-
corrected $M_0$ time series. This analysis aims to evaluate the effectiveness of the moving-average
method in filtering out rainfall-induced interference by assessing the correlation intensity between the
denoised hydrochemical time series and rainfall. At the same time, the analysis verifies potential
temporal linkages between the denoised hydrochemical components and regional seismic moment
release.





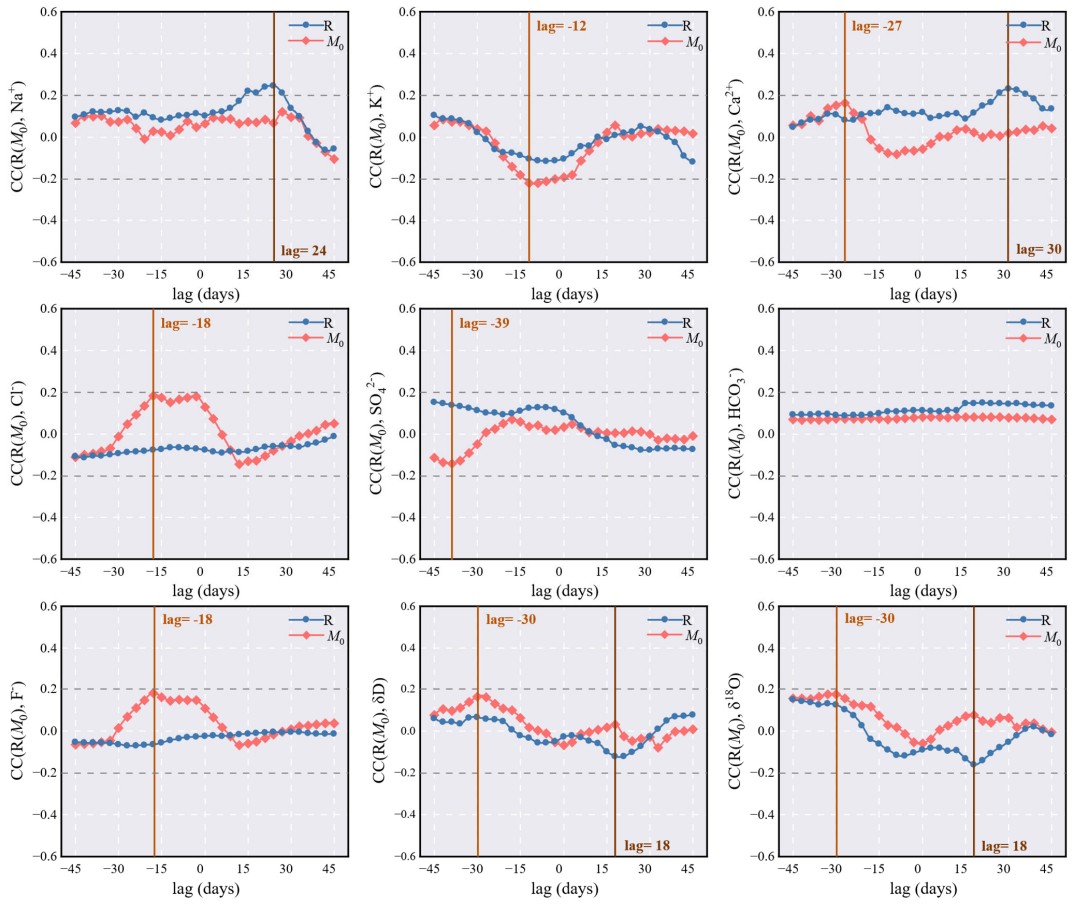

Figure 3. Cross-correlation function analysis of the 15-day moving average time series of hydrochemical components

during stable isotope monitoring, with rainfall and distance-corrected seismic moment.

The results of the cross-correlation analysis (Figure 3) show weak correlations (blue dotted lines)

between rainfall events and the 15-day moving average hydrochemical component time series, with

correlation coefficients approximately within ± 0.2. This result indicates that the moving average

treatment effectively mitigates rainfall-induced noise. Notably, $Na^+$, $Ca^{2+}$, $\delta D$, and $\delta^{18}O$ exhibit minor

response peaks at lags of 18–30 days, suggesting that certain impacts on these components persist 15

to 30 days after rainfall. Similarly, the correlations between the distance-corrected $M_0$ and the denoised



hydrochemical component time series (red dotted lines) remain low (around ± 0.2). However, $K^+$, $Ca^{2+}$,
$Cl^-$, $SO_4^{2-}$, $F^-$, $\delta D$, and $\delta^{18}O$ exhibit weak response peaks at lags of −39 to −12 days, with varying
peak directions for each component. This observation suggests that seismic activity (12–39 days before
seismic moment release) may influence hydrochemical components, causing their concentrations to
fluctuate (either increasing, decreasing, or remaining stable) owing to different geochemical
mechanisms.

**3.4. Detection algorithms**
**3.4.1. BCP analysis**
The correlation analysis results indicate that changes in environmental parameters have little to
no significant influence on denoised hydrochemical components, or their effects were slow processes.
Consequently, these component time series appear to follow specific distribution patterns. However,
since earthquake events are stochastic and hydrochemical anomalies (i.e., change points) emerge
during the pre-earthquake period without a known rupture time, continuous hydrochemical monitoring
data exhibit non-stationary variations. Therefore, BCP analysis is applied to effectively extract
anomalous signals from these component time series.
To detect change points in pre-seismic hydrochemical component time series and verify their
correspondence with earthquakes for forecasting future earthquake occurrences, the BCP analysis
algorithm, which is developed for Earth climate systems (Ruggieri, 2012) to the 15-day moving
average time series of all component concentrations from QJ and WN, is applied to this study. The
analysis produces Bayesian predictive model curves, change point locations, and posterior



probabilities for each component. The posterior probabilities represent the likelihood of change point
occurrences in the predictive models, with probability peaks indicating the most likely timings of
change points.





Figure 4. Anomaly detection results from the Bayesian change point (BCP) analysis applied to hydrochemical
component time series. The black solid line represents the component concentration after 15-day moving averaging.
The green dashed line indicates the forecasting model of the BCP algorithm. The red solid line shows the posterior
probability of change points. Yellow stars mark earthquake events. The false alarm rate (FAR), probability of
detection (POD), and threat score (TS) are evaluation metrics; further details are provided later.

The results show that change points are successfully detected in all component time series from

both thermal springs (Figure 4). For example, before the $M$6.6 earthquake on December 24, 2021, the
posterior probability for a $Ca^{2+}$ change point at QJ was 0.06 at 15 days before the earthquake, while
$SO_4^{2-}$ exhibited a posterior probability of 0.17 at 37 days before the earthquake. Similarly, before the
$M$5.2 earthquake on November 16, 2021, $Cl^-$ at WN showed posterior probabilities for change points
of 0.15 and 0.51 at 3 and 17 days before the earthquake, respectively. Notably, the timing and posterior
probabilities of change points exhibit significant uncertainty, which reflects the complexity of factors
influencing hydrochemical component variations. These factors include inhomogeneity in stress
accumulation, the structural complexity of fault and aquifer systems, modulation by deep gas
degassing, and the mixing effects of multi-source fluids (Skelton et al., 2014; Kim et al., 2019; Hosono
et al., 2020). Although earthquakes result from the coupling of multiple factors, most change points
are identified within 45 days preceding the earthquakes. This observation suggests that component
concentration changes are sensitive to earthquake preparation processes and do occur before
earthquakes, which provides critical empirical support for anomaly detection algorithm models.

The BCP analysis algorithm for anomaly detection also has limitations. Compared with the $Na^+$

detection results at WN, the $Cl^-$ time series produces two false positives for 2022 and misses three



earthquakes for 2023 (Figure 4c, d). This result suggests that the analysis requires a longer time series
and is highly sensitive to prior distribution settings. In practice, identifying the locations and
probabilities of earthquake-induced change points in hydrochemical component time series is
challenging. Larger-amplitude component changes often overshadow smaller-amplitude variations,
which makes the latter difficult to detect and more prone to missed detections or false positives.
Additionally, most BCP methods have a fundamental limitation: they inherently perform retrospective
analysis on complete time series. Specifically, identifying a change point at time $t_i$ relies on data
collected after $t_i$ ($t > t_i$), which makes real-time, forward earthquake forecasting unfeasible with short-
term data sequences (Piersanti et al., 2016).
**3.4.2. Anomaly detection model**

According to the results of the BCP analysis, change-point detection for earthquake forecasting

should be viewed as a supplementary approach. This study enhances the real-time anomaly detection
algorithm for soil radon concentration time series (Piersanti et al., 2016; Soldati et al., 2020) and
applies it to hydrochemical multicomponent time series ($Na^+$, $Cl^-$, $SO_4^{2-}$, $\delta D$, $\delta^{18}O$, etc.). The aim is
to establish an anomaly detection model within a multi-parameter feature space to explore potential
correlations between hydrochemical component variations and major earthquakes. This study modifies
the algorithmic workflow to a backward processing mode and enables real-time forward forecasting.
In the study, evaluation metrics for parameter optimisation are introduced and a seismic response time
threshold parameter, which accounts for local geological conditions is incorporated. The detection
model processes hydrochemical component time series and confirms earthquake catalogues, fitting
optimal parameters based on the evaluation metrics to generate the best anomaly detection parameter



338 combinations for each component. The optimised model performs online, point-by-point data

339 processing for real-time monitoring. When real-time hydrochemical data deviates from the threshold,

340 the model triggers an alarm to predict earthquakes, which enhances forecasting accuracy through

341 multicomponent collaboration (Figure 5). The model improves in three ways: 1) the model

342 incorporates a multi-parameter collaborative verification mechanism that reduces environmental noise

343 interference; 2) the model identifies components with superior anomaly detection performance; 3) the

344 model analyses anomaly intensity based on the number of components detecting anomalies for the

345 same earthquake, thereby improving detection accuracy and reducing false positives and missed alarms.

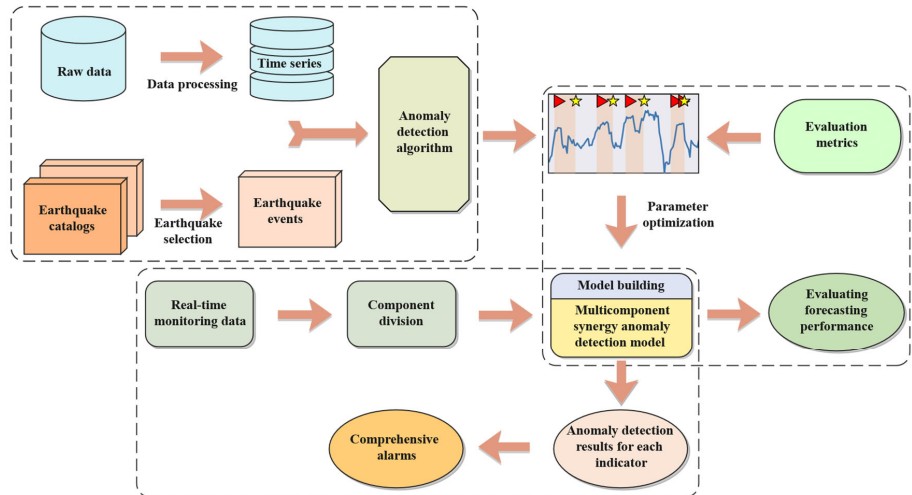

347 Figure 5. Framework of the hydrochemical multicomponent synergy anomaly detection model.

348  The improved detection algorithm procedure (Figure S2) is as follows: Real-time monitoring data

349 for each component from day i are loaded. If the daily value on day i−p2 exceeds p1 times the 15-day

350 moving average on day i−p2−1, and simultaneously, the 15-day moving average on day i surpasses p3

351 times that on day i−p2, the system triggers an alarm on day i. This alarm is considered a successful





early warning if an $M \geq 4$ earthquake occurs within $I + T_{th}$. After an earthquake, new parameters
p1' = p4 × p1 and p3' = p4 × p3 are used for a period of p5^M (where $M$ is the magnitude). If
subsequent earthquakes occur within this period, the post-earthquake time is calculated based on the
maximum magnitude. If no $M \geq 4$ earthquakes occur during this interval, parameters p1 and p3
automatically revert to their initial values. The algorithm incorporates five adjustable parameters (p1–
p5) and a seismic response time threshold ($T_{th}$), with p1, p3, and $T_{th}$ being key parameters.

When thermal water is subjected to external disturbances (e.g., contamination or anthropogenic

inputs), particularly the dissolution of a single compound, variations in hydrochemical ion
concentrations generally follow the charge balance principle, often resulting in synchronous changes
in paired cations and anions. To minimise the impact of uncertain interference and improve program
efficiency, reliable warning signals should be defined by concurrent alarms from at least three
hydrochemical components. The intensity of the anomaly increases with the number of components
triggering simultaneous alarms.
**3.4.3. Evaluation metrics**

Given that earthquake forecasting research focuses on evaluating algorithms' ability to identify

low-probability earthquake events, this study employs four evaluation metrics based on the number of
correct alarms (NA), false alarms (NB), and missed alarms (NC).
False alarm rate (FAR): This measures the proportion of non-earthquake events incorrectly classified
as earthquake events, relative to the total number of warning instances.

$$FAR = NB/(NA + NB) \tag{7}$$

Missed alarm rate (MAR): The proportion of earthquakes that are not detected relative to the total



number of earthquake events, indicating the risk of failing to identify such events.

$$MAR = NC/(NA + NC) \qquad (8)$$

Probability of detection (POD): The proportion of correctly identified earthquake events out of all
earthquake events, assessing the model's ability to detect these events.

$$POD = NA/(NA + NC) \qquad (9)$$

Threat score (TS): This reflects the accuracy of the forecast, ranging from 0 (complete mismatch) to 1
(perfect match) with actual events.

$$TS = NA/(NA + NB + NC) \qquad (10)$$

This metric system allows for a more accurate evaluation of the model's forecasting performance in
handling imbalanced data through multi-dimensional quantitative analysis.

**4. Results and discussion**
**4.1. Hydrochemistry**

The average water temperature at QJ is approximately 60°C, with a pH of 7.5 and an EC of 1148

μS/cm, while WN has a higher temperature of 80°C, a pH of 7.9, and a lower EC of 579 μS/cm. Both
QJ (sandstone) and WN (mylonite, gneiss, etc.) exhibit similar hydrochemical types ($HCO_3$-Na),
owing to the comparable lithology of the surrounding rocks. The $\delta^{18}O$ values at QJ range from −13.19‰
to −11.81‰, while the δD values range from −94.93‰ to −90.59‰. At WN, the $\delta^{18}O$ values range
from −13.22‰ to −12.01‰, and the δD values range from −91.26‰ to −88.09‰. The narrow
fluctuation range of stable isotopes in both thermal springs, coupled with their proximity to the local
and global meteoric water lines (Figure S3), suggests that the thermal spring water originates from



atmospheric precipitation. Overall, the two springs exhibit similar hydrochemical characteristics,
which minimises the impact of compositional differences on the evaluation of algorithm effectiveness
across the different springs. For detailed hydrochemical ion concentrations and isotope values, please
refer to the supplementary materials.

**4.2. Model parameters**
**4.2.1. Seismic response time threshold**

The anomaly detection model in this study establishes forecasting rules based on the temporal

correlation between precursor anomalies and earthquake events. The seismic response time threshold
($T_{th}$) plays a key role in determining both forecasting performance and practical value. $T_{th}$ is defined
as the maximum allowable time interval between anomaly detection and earthquake occurrence. This
threshold is a critical parameter that balances accuracy and timeliness. Increasing $T_{th}$ expands the
monitoring window and captures more potentially correlated abnormal signals, but it significantly
reduces the time resolution of forecasting. Conversely, decreasing $T_{th}$ improves temporal precision but
may risk omitting valid precursor signals owing to shorter observation periods.

To improve the accuracy of the anomaly detection model in predicting earthquake timing, the

nonlinear effects of $T_{th}$ on predictive performance are systematically explored via an incremental
increase of $T_{th}$ from 5 to 70 days in 5-day steps. This increase identifies key inflection points during
threshold optimisation (Figure S4). As $T_{th}$ increases from 5 to 45 days, model performance improves
considerably, with both TS and POD rising rapidly, while FAR gradually decreases. This improvement
results from the extended monitoring windows, which better capture the association between





anomalies and seismic activities. Notably, the evaluation metrics reveal a turning point at the 45-day
threshold. Beyond 45 days, the trends in TS, POD, and FAR plateaus, with minimal variation. This
result is consistent with the finding that maximum cross-correlations between $M_0$ and hydrochemical
components ($Cl^-$, $SO_4^{2-}$, $\delta D$) occur within 45 days before the earthquake (Figure 3) and that BCP
analysis detects most change points emerging within 45 days of pre-earthquake events (Figure 4).
Collectively, these results jointly define 45 days as the optimal response time threshold for
hydrochemical precursors to seismic activities in the study area.
**4.2.2. Free parameters**
The parameter optimisation process involves quantitatively aligning observed hydrochemical
data with seismic precursor anomalies. Among the five adjustable parameters (p1–p5) in the detection
model, the key regulatory parameters p1 and p3 represent multiples of the sliding window values. This
study focuses on p1 and p3 to examine the influence of the optimisation of these parameters on the
performance of the anomaly detection model. For optimisation involving ion concentration data, the
model applies parameter values ranging from 1.00 to 1.20 in steps of 0.01. For optimisation involving
isotopic data, which exhibit minor fluctuations, the model applies parameter values ranging from 0.985
to 1.015, with a step increment of 0.001. Model performance is then evaluated using TS. Figure 6
shows the variations in the TS under different values of p1 and p3. When p1 and p3 are small, the
model becomes overly sensitive to background noise, detecting more non-seismic signals. This effect
leads to an increase in the FAR and a decrease in TS. As p1 and p3 increase, TS improves. However,
when the parameters become excessively large, surpassing the actual seismic anomaly thresholds, the
MAR rises sharply, which causes TS to drop below 0.35. The optimal parameter combinations for each



hydrochemical component are identified at the TS peak inflection points (marked by yellow circles).
According to this method, the complete set of model parameters for all hydrochemical components at
QJ and WN is provided in Table S2.

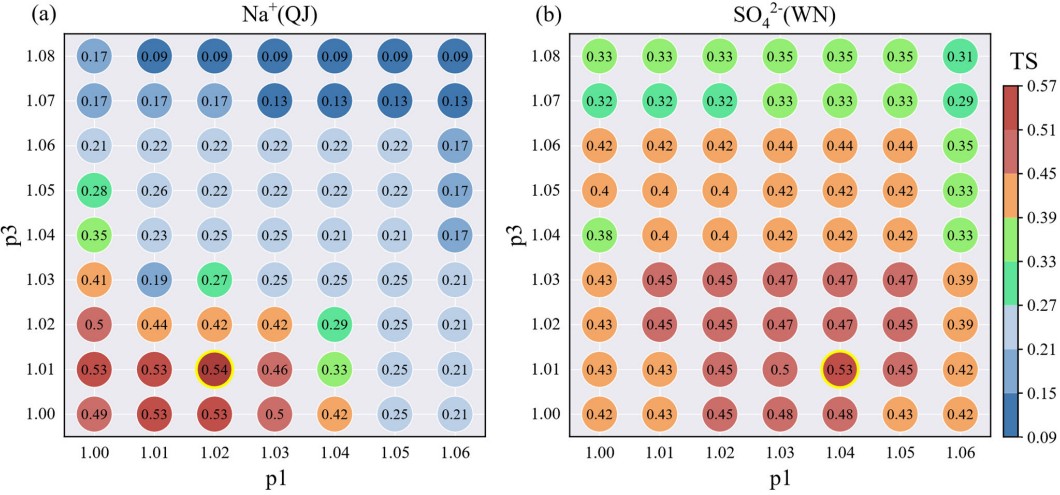


Figure 6. Effect of parameters on model performance. TS varies with changes in p1 and p3 (the main free
parameters p1 and p3 are used as examples). The yellow circle highlights the TS value corresponding to the optimal
combination of p1 and p3.

The parameter optimisation results reveal notable differences in the optimal p1 and p3

combinations for different hydrochemical components. Hydrochemical anomalies preceding
earthquakes in many regions are typically caused by the combined action of multiple mechanisms
(Skelton et al., 2014; Kim et al., 2019; Hosono & Masaki, 2020). Furthermore, owing to significant
variations in the geochemical behavior of different hydrochemical components, components within
the same thermal spring often exhibit diverse response patterns to the same earthquake. These patterns
may include asynchronous variations (increase/decrease/stability) and considerable discrepancies in
the magnitude of change (Shi et al., 2020; Wang et al., 2021; Tian et al., 2023). Therefore, optimising



parameter combinations to create customised anomaly detection models for specific hydrochemical
components at designated observation points is the key approach in this study to enhance the model's
ability to detect seismic precursor information.

**4.3. Evaluation of forecasting performance**

Figures 7 and 8 present the 15-day moving average time series of hydrochemical components,

anomaly detection results, and earthquake events for the anomaly detection model at QJ and WN. For
each component, the model successfully identifies varying numbers of pre-earthquake anomalies and
triggered warnings. The model activates comprehensive alarms when anomalies are detected in three
or more components simultaneously. At QJ, the model provides 21 effective warnings for 22
earthquake events (POD = 0.95), with 8 false alarms (FAR = 0.28) and a TS of 0.70. At WN, the model
generates 10 accurate warnings for 12 events (POD = 0.83), 5 false alarms (FAR = 0.33), and a TS of
0.59. Compared with the single-component anomaly detection results, the multi-component joint
warning results exhibit higher TS values (Figures 7, 8, 9). This observation demonstrates that
multicomponent collaboration mitigates the effects of geochemical behavior differences among
components, reduces environmental interference on individual ions/ion pairs, and consequently
enhances the accuracy of the anomaly detection model. Zhu et al. (2024) comprehensively evaluated
the anomaly detection performance of several machine learning algorithms using 2.5 years of
hydrochemical data from the southeast coast of China. The best-performing local outlier factor
algorithm achieved an R-score of about 0.6, POD of about 0.7, and FAR of about 0.15. The improved
anomaly detection model demonstrates comparable performance, which confirms its effectiveness.



The results from the anomaly detection model and BCP analysis are mutually corroborative; however,
the anomaly detection model exhibits superior sensitivity in processing nonlinear time series data.
Taking QJ as an example, the model achieves POD values of 0.70 and 0.59 for $Ca^{2+}$ and $SO_4^{2-}$ detection
results, respectively (Figures 4 and 7), which represents significant improvements over the BCP
analysis results (0.50 and 0.41). The model is also capable of accurately detecting subtle anomalies
that the BCP analysis may miss.

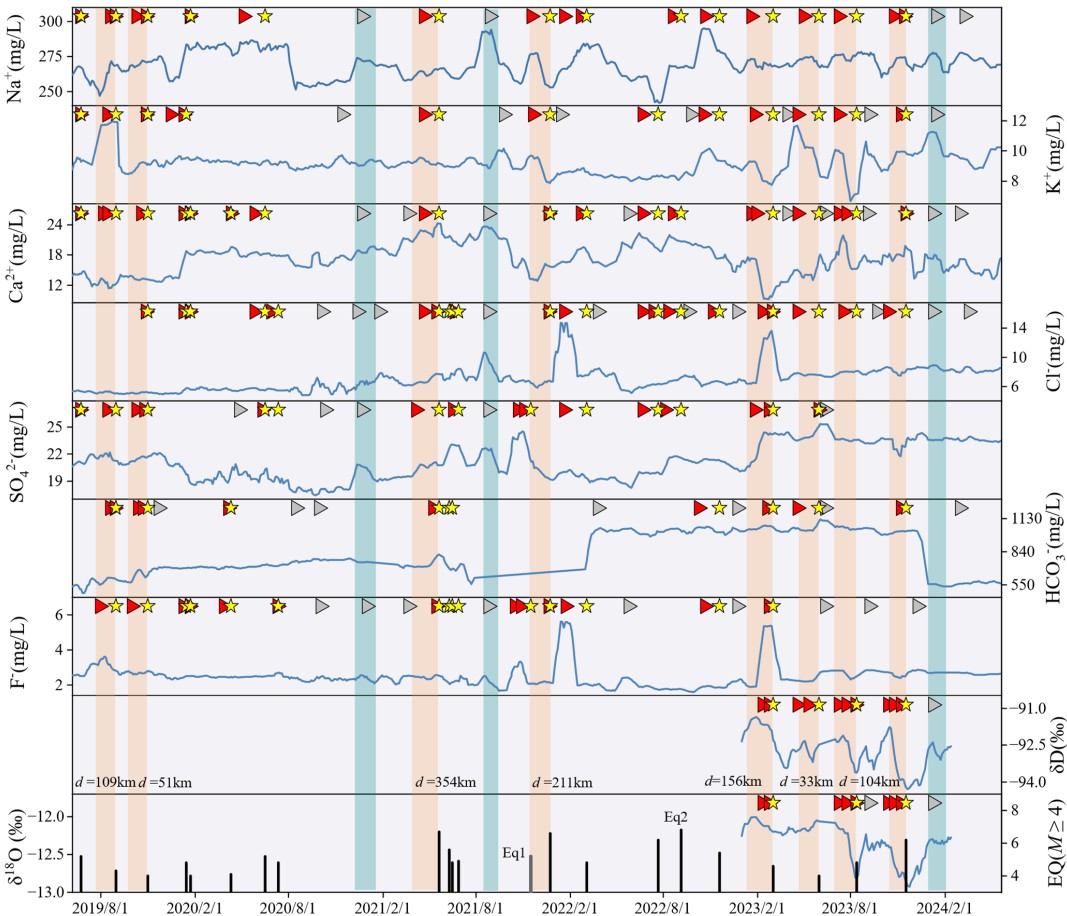


Figure 7. Results of the anomaly detection model applied to hydrochemical component time series from Qujiang
spring. The blue curve represents the hydrochemical component time series after a 15-day moving average. Red and





gray triangles indicate accurate alarms and false alarms, respectively. Yellow stars mark successfully reported
earthquakes. Black and gray vertical bars show detected and missed earthquakes based on the algorithm's
comprehensive alarm (triggered by $\geq 3$ components), respectively. Orange-red boxes highlight synchronous
successful alarms triggered by six or more components. Grayish-blue boxes mark synchronous false alarms triggered
by five or more components.
Owing to variations in the geochemical behaviors of hydrochemical components, their response
patterns and magnitudes to earthquakes differ. Although the mechanisms behind these differences have
not yet reached academic consensus, this study aims to identify effective strong earthquake prediction
indicators applicable to the study area through anomaly detection model results. A comparison of the
TS values of each component's warning results in QJ and WN (Figure 9) reveals that in the two thermal
springs of the study area, the TS values for $Na^+$, $Ca^{2+}$, $Cl^-$, $SO_4^{2-}$, $\delta D$, and $\delta^{18}O$ detection (around 0.50)
are relatively high. This observation suggests that these components can serve as sensitive indicators
for strong earthquake forecasting in the study area. In general, QJ in the study area exhibits a more
sensitive response to earthquakes. In addition, the anomalies are categorised into multiple consecutive
anomalies and single anomalies (Figures 7 and 8). This phenomenon is more pronounced in the stable
isotope time series, likely because isotopic changes are more sensitive and tend to trigger multiple
warning signals before an earthquake.



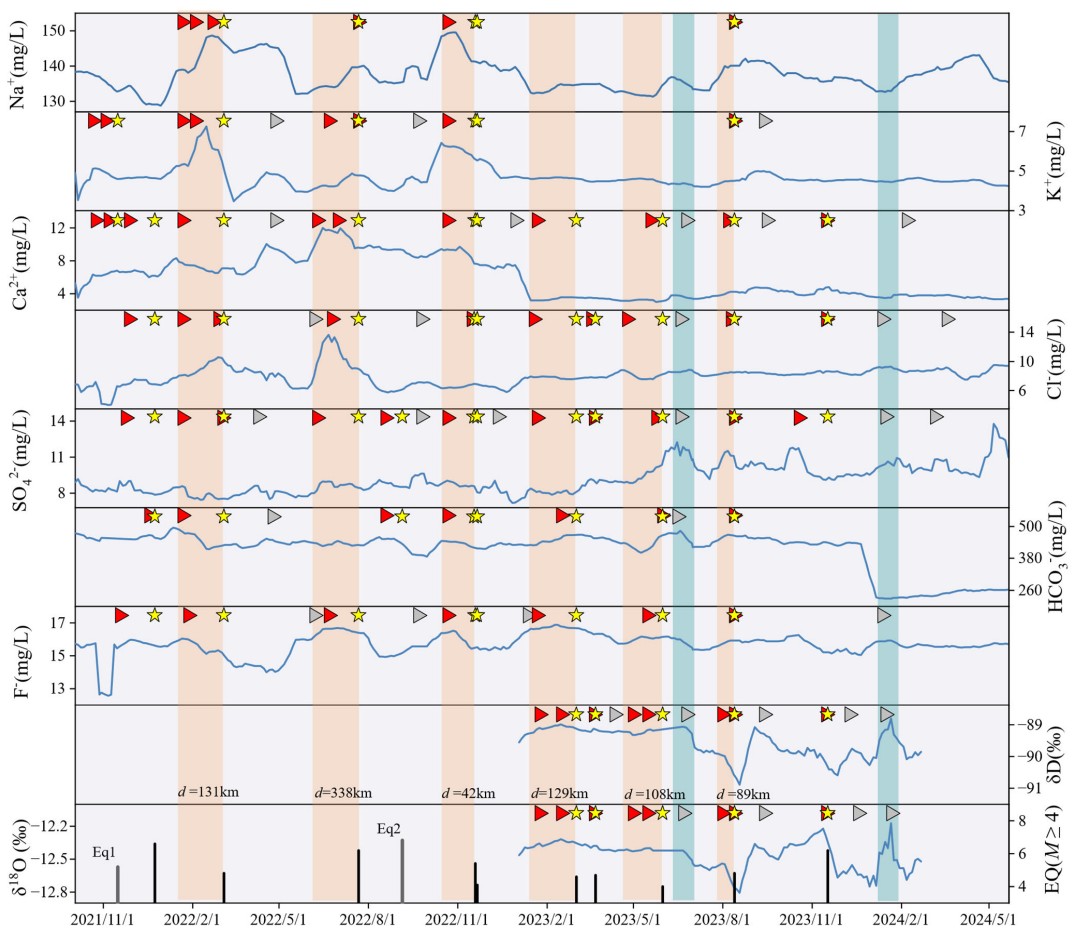

Figure 8. Results of the anomaly detection model applied to hydrochemical component time series from Wana

spring. The same notes as above for Figure 7.

Among the earthquakes for which the earthquake preparation zone covers both thermal springs,

only two earthquakes (represented by gray vertical bars in Figures 7 and 8) fail to induce

multicomponent anomalies prior to the earthquake. Earthquake Eq1 causes no synchronous anomalies

at either spring, which suggests that Eq1 has a limited impact on regional tectonic activity. For Eq2

(epicentral distance > 600 km), WN shows no alarm response, while QJ reacts effectively. This

discrepancy is likely related to WN's location on the eastern boundary of the SYB, where stress



accumulation mainly affects QJ, which is also located on the eastern border. The muted response in
WN likely results from the blocking effects of the RRF (Li et al., 2024; Shao et al., 2024). The similar
abnormal response sensitivity of different springs to the same earthquake demonstrates regional-scale
hydrochemical impacts from earthquake preparation and confirms the stable and reliable performance
of the anomaly detection model.

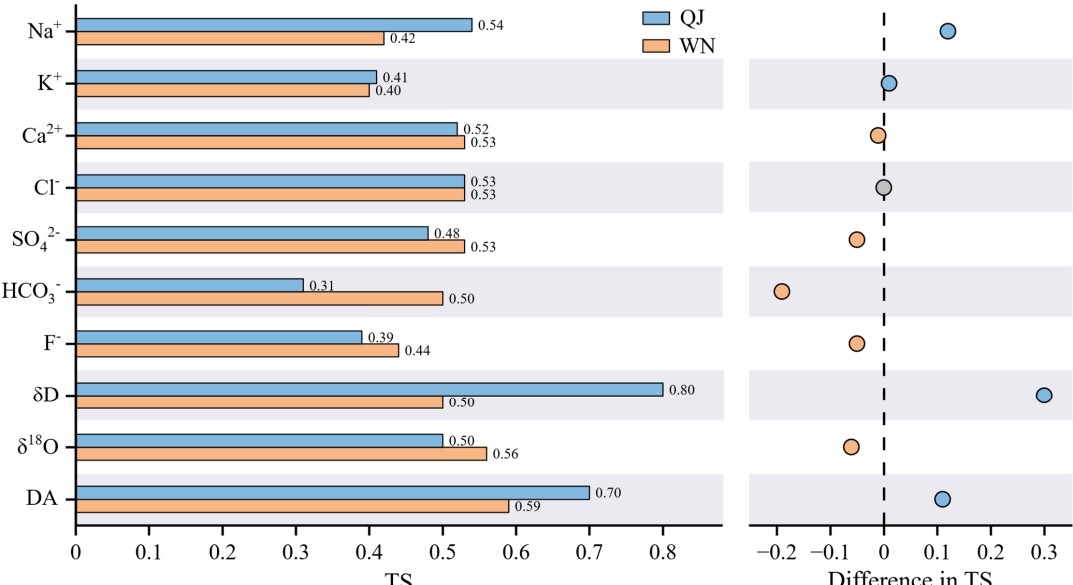


Figure 9. TS values of anomaly detection model results for hydrochemical components. DA denotes comprehensive
alarms triggered by the model.

In Figures 7 and 8, the orange-red boxes represent model results of successfully predicted

earthquakes, identified through synchronised anomalies in six or more hydrochemical components.
The width of the boxes, which indicates the interval between the appearance of anomalies and
earthquake occurrence, shows no clear correlation with magnitude or epicentral distance. This
observation underscores the complex dynamic mechanisms and regional structural differences





involved in the earthquake preparation process, with anomalies often arising from the combined effects
of multiple mechanisms (Thomas, 1988). Spatially, the number of hydrochemical components
exhibiting synchronous anomalies correlates with earthquake magnitude and epicentral distance.
Earthquakes that induce synchronous anomalies in six or more hydrochemical components have
epicentral distances within 150 km for earthquakes with magnitudes less than 6.0 ($M < 6.0$), while this
distance extends to approximately 450 km for earthquakes with magnitudes greater than or equal to
6.0 ($M \geq 6.0$). Although it is difficult to quantify the exact impact of magnitude and distance on the
number of components exhibiting synchronous anomalies, as magnitude increases or distance
decreases, the number of components with synchronous anomalies detected by the model tends to
increase. This trend aligns with the positive correlation between the scale of earthquake energy release
and the number of anomalies, as confirmed by the hydrochemical monitoring results (Li et al., 2022).
Therefore, a significant relationship exists between the temporal variation of hydrochemical
components and earthquakes in the study area. The number of components exhibiting synchronous
anomalies can be used as an effective criterion for determining alarm intensity, with higher intensity
generally corresponding to larger earthquake magnitudes or shorter epicentral distances.

Furthermore, this study reveals that hot springs closer to the epicenter tend to exhibit a greater

number of components with synchronous anomalies during the same earthquake. Pre-earthquake
hydrochemical anomalies generally manifest on a regional scale, which means different thermal
springs can not only validate each other in terms of anomaly timing for forecasting purposes but also
help identify the closest springs to the epicenter based on the number of synchronous anomalous
components. This approach aids in defining potential earthquake preparation zones. According to this



approach, a dense thermal spring monitoring network provides more opportunities for spatial
earthquake forecasting.

**4.4. Limitations and prospects**

This study focuses on evaluating the performance of anomaly detection models in predicting the

timing of earthquakes with magnitudes ≥ 4. One potential cause of false alarms could be anomalous
fluctuations in hydrochemical components triggered by seismic activities with magnitudes < 4 in areas
near thermal springs. Four days after the second synchronised false alarm involving five components
(Figure 7, grayish-blue boxes), an $M$2.6 earthquake occurred 3 km from QJ. This occurrence suggests
that high-frequency false alarms may not solely result from non-seismic fluid anomalies, but could
also reflect the model's limited ability to distinguish anomalies caused by microseisms. According to
this finding, it is recommended to establish observation station networks and optimise algorithms to
enable hierarchical alarm systems. Approximately 30 days after the last multicomponent synchronised
false alarms at the two thermal springs (Figures 7 and 8), an $M$4.1 earthquake occurred, with an
epicenter located outside the radius of the earthquake preparation zone. Current earthquake screening
criteria assume an isotropic underground structure; however, the algorithm requires tailored
optimisation based on the specific geological background in practical applications. Additionally, the
model demonstrates limited adaptability to changes in data trends, highlighting the need for periodic
parameter re-optimisation. While the model is constructed using major elements and stable isotopic
indicators in thermal waters, future research should also consider the potential associations between
other hydrochemical components, such as trace elements, and seismic activity.




## 5. Conclusions

A multicomponent synergistic anomaly detection model is developed using five years of
continuous hydrochemical monitoring data to enable real-time forecasting of $M \geq 4$ earthquakes in the
study area. Model parameters are optimised for each component, and their impact on anomaly
detection performance is evaluated to identify applicable hydrochemical indicators for strong
earthquake forecasting. The results of the multicomponent synergy anomaly detection reveal a clear
connection between hydrochemical variations and seismic activity, offer valuable insights, and
establish a new paradigm for precursor identification in earthquake forecasting. The main findings are
summarised as follows:
1. A 45-day response time threshold for hydrochemical components to $M \geq 4$ earthquakes is
established as the optimal period for capturing key hydrochemical precursors for short-term
earthquake forecasting. Tailored model parameters for specific hydrochemical components
account for their differences in response characteristics to seismic activity and significantly
enhance the model's performance and adaptability.
2. The anomaly detection model demonstrates reliable real-time anomaly detection capabilities and
identifies Na, $Ca^{2+}$, $Cl^-$, $SO_4^{2-}$, δD, and $δ^{18}O$ as effective indicators for strong earthquake
forecasting, with δD and $δ^{18}O$ exhibiting higher sensitivity to seismic activity.
3. The newly proposed multi-parameter synergy alarm mechanism for hydrochemistry overcomes the
limitations of single-parameter methods and significantly improves the model's overall
performance in earthquake forecasting. The number of hydrochemical components with



synchronous anomalies provides a reliable criterion for determining alarm intensity, with higher
intensity typically correlating to larger earthquake magnitudes or shorter epicentral distances. A
dense thermal spring monitoring network can facilitate cross-verification across multiple sites for
time-based forecasting and offer enhanced capabilities for spatial forecasting.

**Acknowledgments**
This work was jointly supported by National Key Research and Development Project (No.
2023YFC3012005-1), National Natural Science Foundation of China (No. 42073063), Special Fund
of the Institute of Earthquake Forecasting, China Earthquake Administration (No. CEAIEF20230301),
Deep Earth Probe and Mineral Resources Exploration National Science and Technology Major Project
(No. 2024ZD1000503). This work is a contribution to IGCP Project 724.

**Declaration of interest**
The authors declare that they have no known competing financial interests or personal
relationships that could have appeared to influence the work reported in this paper.

**Data availability**
The continuous monitoring data from thermal springs can be found at Mendeley Data, Version 1
(https://10.17632/xkd75cyfmb.1).



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
