# Peer review of "Decoding multicomponent hydrochemical anomalies: A synergy"

_EGUsphere, 2025_

## Author Comment (AC1)

**RESPONSES TO REVIEWER ONE'S COMMENTS**

We would like to express our sincere appreciation for your valuable comments and suggestions on our manuscript. We have carefully considered the comments and have revised the manuscript accordingly. The comments are laid out below in italicized font. Our response is given in normal font and changes/additions to the manuscript are given in the blue text.

**# *General comments:**

1. *#Abstract: It would be worth rephrasing to make the message clear and better reflect the key findings and the value of this study.*

**Response:** Thank you for this constructive suggestion. We agree entirely that enhancing the clarity and impact of our findings will strengthen the paper. We have revised the Abstract to better underscore our key findings and the value of this study as follows:

The intersection of the Xiaojiang Fault and the Red River Fault at the southeastern margin of the Tibetan Plateau experiences intense tectonic activity. At this intersection, frequent earthquakes have induced hydrochemical variations in thermal springs. In this study, bayesian change point analysis is applied, and a multicomponent synergy anomaly detection model is developed using five years of monitoring data (2019–2024) from two thermal springs in the region to achieve real-time forecasting of occurrence timing for $M \geq 4$ earthquakes. A 45-day response time threshold is established as the optimal period for capturing key hydrochemical precursors to $M \geq 4$ earthquakes. Parameters are optimized for individual components based on their distinct geochemical responses to seismic stress, thereby significantly enhancing the model's

performance and adaptability. The model shows reliable real-time detection capabilities, with probability of detection (POD) ranging from 0.83 to 0.95 and threat score (TS) between 0.59 and 0.70. It identified pre-earthquake high-value anomalies in $Na^+$, $Ca^{2+}$, $Cl^-$, $SO_4^{2-}$, $\delta D$, and $\delta^{18}O$, with TS $\geq$ 0.50, which can serve as sensitive indicators for strong earthquake forecasting. The multicomponent synergy alarm mechanism for hydrochemistry overcomes the limitations of single-parameter methods and improves overall forecasting performance. The number of hydrochemical components with synchronous anomalies serves as a reliable criterion for determining alarm intensity, with higher intensity typically correlating with larger earthquake magnitudes or shorter epicentral distances. The model can be universally applied to hot spring monitoring across diverse tectonic regions through targeted parameter optimization, offering an attempt at a method to advancing earthquake forecasting.

2. *#Introduction: The Introduction is mostly well written. However, some minor issues should be state clearer and some relevant references are missing. Please see minor comments below.*

**Response:** We appreciate your careful reading and valuable suggestions for improvement. We have carefully addressed all minor comments provided below to enhance clarity where needed and have added the relevant references as requested. The revised Introduction now incorporates these improvements.

3. *#Method and data:*

*a) Some results are presented and discussed in Section 3.3 and 3.4, which makes the structure unclear. The authors are suggested to reorganize some contents in 3.3 and 3.4, and move them into results and discuss them accordingly.*

*b) Also, some contents in this section are too lengthy. The authors are suggested to simplify some of the method (for example, the introduction of limitations of BCP*

*method could go to later section or Supplementary information).*

**Response:** Thank you for your thoughtful suggestions regarding the organization and conciseness of the manuscript. We agree that the structure could be improved for better clarity and flow. In response to point (a), we will reorganize the content in Sections 3.3 and 3.4 by moving appropriate parts to the Results and discussion sections, respectively, to more clearly distinguish between experimental findings and their interpretation. For point (b), we will simplify the Methods section by relocating the detailed limitations of the BCP method to the Discussion section, as suggested, to maintain focus on the core methodology. We will incorporate these changes in the revised manuscript.

4.  *#Results and discussion: The results presented here are convincing; however, some lack in-depth discussion, causing some implications of the study to be obscured. It is recommended that the authors further discuss how some of these findings could be applied to other tectonically active regions around the world.*

**Response:** We thank the reviewer for their positive assessment of our results and their valuable suggestion regarding the discussion. We agree that further elaboration on the broader implications would strengthen the manuscript. In response, we have significantly expanded the Results and discussion section to explicitly discuss how these findings could be applied to other tectonically active regions globally.

We added the following content to the last part of the 4.4. Limitations and prospects section:

Owing to significant differences in hydrogeological settings, tectonic activity, and the current limitations in quantitatively modeling geothermal water circulation under specific geological conditions, a universal set of model parameters applicable across all hot springs within even the same tectonic region cannot be established. This highlights the necessity of anomaly detection model, which involves optimizing parameters specifically for individual hot springs based on their unique pre-seismic responses in

different hydrochemical components. The model aims to leverage the inherent differences among these hydrochemical components, integrating them to enhance forecasting efficacy. Crucially, this methodological framework is transferable. For application in other tectonic regions, the model can be adapted by similarly optimizing the parameter combinations for the target hot spring(s) based on their specific hydrochemical components. This addresses the challenge posed by varying tectonic and hydrogeological conditions leading to divergent hydrochemical behaviors. By enabling the application of the model to hot spring monitoring in specific regions through this targeted parameter optimization, the model provides an attempt at a method to advancing earthquake forecasting.

**# Specific comments:**

1. #Lines 26-27 Please specify how these isotopes changes before earthquake.

**Response:** We appreciate your comment. Indeed, directly observable significant changes in hydrochemical components before earthquakes are not commonly recorded across most seismic events and have been reported primarily in certain representative cases, often marked by high abnormal values (i.e., increased concentrations) (Skelton et al., 2014; Zhang et al., 2021; Gori and Barberio, 2022; Yan et al., 2022; Yakupoğlu et al., 2025). More frequently, precursory signals are subtle, which highlights the need for sensitive detection methods.

In this study, we employed a model designed to capture both conspicuous and subtle anomalies, with the aim of identifying more short-term precursory signals. As continuous hydrochemical monitoring continues to develop, there is a growing need to identify reliable hydrochemical indicators for practical earthquake forecasting applications, this is a key motivation of our work.

Based on existing understanding of pre-earthquake anomalies, our model was configured under the assumption that ion and isotope concentrations typically exhibit sustained high values prior to earthquakes. Specifically, the model triggers an alarm

when the following conditions are met: If the daily value on day i−p2 exceeds p1 times the 15-day moving average on day i−p2−1, and simultaneously, the 15-day moving average on day i surpasses p3 times that on day i−p2, the system triggers an alarm on day i.

This logic effectively targets sustained increases in concentration, ensuring that both abrupt rises and gradual accumulations of ions/isotopes are captured. Therefore, the pre-earthquake changes in these hydrochemical components identified by the model are all sustained high-value anomalies.

Based on the above statement, we have added more details in Lines 26-27:

The model identified pre-earthquake high-value anomalies in $Na^+$, $Ca^{2+}$, $Cl^-$, $SO_4^{2-}$, $\delta D$, and $\delta^{18}O$, with a threat score (TS) value exceeding 0.50, which can serve as sensitive indicators for strong earthquake forecasting.

**References:**

Gori, F., Barberio, M. D.: Hydrogeochemical changes before and during the 2019 Benevento seismic swarm in central-southern Italy, Journal of Hydrology, 604, https://doi.org/10.1016/j.jhydrol.2021.127250, 2022.

Skelton, A., Andrén, M., Kristmannsdóttir, H., Stockmann, G., Mörth, C. M., Sveinbjörnsdóttir, Á., Jónsson, S., Sturkell, E., Guðrúnardóttir, H. R., Hjartarson, H., Siegmund, H., and Kockum, I.: Changes in groundwater chemistry before two consecutive earthquakes in Iceland, Nature Geoscience, 7(10), 752-756, https://doi.org/10.1038/ngeo2250, 2014.

Yakupoğlu, N., Sabuncu, A., Erbil, C., Kırkan E., Çetin H., and İnan S.: Pre-earthquake hydrogeochemical anomalies in spring waters: two distinctive cases from western Türkiye, Journal of Hydrology, 662, https://doi.org/10.1016/j.jhydrol.2025.133920, 2025.

Yan, Y., Zhou, X., Liao, L., Tian, J., Li, Y., Shi, Z., Liu, F., and Ouyang, S.: Hydrogeochemical Characteristic of Geothermal Water and Precursory Anomalies along the Xianshuihe Fault Zone, Southwestern China, water, 14(4),

https://doi.org/10.3390/w14040550, 2022.

Zhang, L., Guo, L., Zhou, X., Yang, Y., Shi, D., and Liu, Y.: Temporal variations in stable isotopes and synchronous earthquake-related changes in hot springs, Journal of Hydrology, 599, https://doi.org/10.1016/j.jhydrol.2021.126316, 2021.

*2.  #Line 79 Please add relevant references for this statement.*

**Response:** We added two references to support this statement on line 79:

The hydrochemical components (e.g., $Na^+$, $Cl^-$, $SO_4^{2-}$) of thermal springs tend to exhibit high stability, rapid upward migration, and limited susceptibility to environmental interference (Luo et al., 2023; Yakupoğlu et al., 2025).

**References:**

Luo, Z., Zhou, X., He, M., Liang, J., Li, J., Dong, J., Tian, J., Yan, Y., Li, Y., Liu, F., Ouyang, S., Liu, K., Yao, B., Wang, Y., and Zeng, Z.: Earthquakes evoked by lower crustal flow: Evidence from hot spring geochemistry in Lijiang-Xiaojinhe fault, Journal of Hydrology, 619, https://doi.org/10.1016/j.jhydrol.2023.129334, 2023.

Yakupoğlu, N., Sabuncu, A., Erbil, C., Kırkan E., Çetin H., and İnan S.: Pre-earthquake hydrogeochemical anomalies in spring waters: two distinctive cases from western Türkiye, Journal of Hydrology, 662, https://doi.org/10.1016/j.jhydrol.2025.133920, 2025.

*3.  #Line 89 what are the common machine learning algorithms.*

**Response:** We sincerely thank the reviewer for identifying the lack of clarity in this statement. Given the limited sample size inherent to earthquake precursor studies—owing to short monitoring histories and low seismic occurrence rates—data availability often precludes data-intensive deep learning architectures. Therefore, 'common machine learning algorithms' herein refer to widely adopted methods in hydrochemical anomaly detection, including Isolation Forest, Local Outlier Factor, Autoencoder,

among others. We have revised the text to explicitly specify the algorithms referenced for clarity. The updated sentence on line 89 now reads:

Existing studies have demonstrated the effectiveness of widely adopted machine learning algorithms (e.g., Isolation Forest, Local Outlier Factor, and Autoencoder) in identifying abnormal periods in hydrochemical data while also emphasising the need for scenario-specific optimisation of key indicators (Zhu et al., 2024).

4. *#Line 172 Please provide the references for this equation and explain the meaning of each parameter.*

**Response:** Thank you for your feedback. We confirm the validity of this equation, with reference and parameter clarifications are provided in lines 171-174:

To ensure data accuracy, cation–anion balance error tests were performed for each sample, with all ionic deviations kept within ± 5%. The ion balance error (Appelo and Postma, 2004) is calculated as below:

$$ib(\%) = \frac{\sum cations - \sum anions}{\sum cations + \sum anions} \times 100 \tag{1}$$

where $\sum cations$ represents the sum of cation concentrations (in milliequivalents per liter, meq/L), and $\sum anions$ represents the sum of anion concentrations.

**References:**

Appelo, C.A.J., and Postma, D.: Geochemistry, Groundwater and Pollution (2nd ed.), A.A. Balkema Publishers, Leiden, 17pp, ISBN04 1536 428 0, 2004.

5. *#Line 186-187 Ambiguous. Consider rephrasing it to: '22 earthquakes with M ≥ 4'.*

**Response:** We agree with the reviewer's point and will revise the sentence accordingly for clarity. The revised part is:

The QJ site was within the preparation zones of 22 earthquakes with $M \geq 4$ during its monitoring period (2019/06/01–2024/05/21), while the WN site was within the

preparation zones of 12 earthquakes with $M \geq 4$ during its observation period (2021/10/03–2024/05/21) (Table S1).

6. *#Line 203 Please explain why you chose ω=1. Have you conducted a sensitivity analysis?*

**Response:** Thank you for raising this important point. In this study, ω serves as a coefficient in the seismic moment distance attenuation correction, which was introduced to optimize and quantify the potential correlation between seismic moment and hydrochemical component. Based on previous research focused on radon (Rn), values of ω = 1.3 and ω = 3 have been commonly adopted (Piersanti et al., 2016). We conducted a sensitivity analysis by testing multiple ω values (including 0, 0.5, 1, 1.5, 2, 2.5, and 3) and observed that the correlation peak consistently emerged within the same lag range across all ω settings. Among these, ω = 1 produced the pronounced correlation result (Figure S5).

We apologize for not providing a more detailed explanation in the original manuscript. Since ω is not a critical parameter in the model and does not affect the prediction target of M ≥ 4 earthquakes, we did not include an extensive sensitivity analysis. However, in response to your comment, we will be happy to add these details in the Supplementary Materials to improve the clarity of our approach.

The supplementary information is as follows:

[Figure]

**Figure S5.** Cross-correlation function analysis of the 15-day moving average time

series of hydrochemical components and distance-corrected seismic moment for multiple ω values (including 0, 0.5, 1, 1.5, 2, 2.5, and 3).

Figure S5 presents the cross-correlation analysis results between $M_0$ with different ω values and $K^+$, $F^-$, and δD, which represent the cations, anions, and water isotopes with prominent correlations, respectively. The ω values range from 0 to 3 with a step increment of 0.5. As ω varies, the peak of the lag time remains stable, indicating that relationships exist between $M_0$ and hydrochemical components at specific lag times. When ω takes values of 0, 0.5, and 1, the cross-correlations are relatively significant and exhibit minimal difference. Therefore, considering the practical physical significance, this study selects a ω value of 1.

**References:**

Piersanti, A., Cannelli, V., and Galli, G.: The Pollino 2012 seismic sequence: clues from continuous radon monitoring, Solid Earth, 7(5), 1303-1316, https://doi.org/10.5194/se-7-1303-2016, 2016.

*7.   #Line 238 Please explain why a 15-day backward moving average is applied.*

**Response:** We thank the reviewer for their comment regarding the use of the 15-day backward moving average. This data processing step is fundamental in earthquake-related fluid geochemistry for the following reasons:

Noise filtration and signal preservation: The primary purpose is to effectively filter out short-term, high-frequency noise, predominantly caused by rainfall infiltration and dilution effects, which create sharp spikes in the data. Ambient temperature and atmospheric pressure at the spring outlet are ignored because they have a negligible effect on the hydrochemistry. Simultaneously, this window size is optimal for preserving medium-to-long-term trends that are more likely to be associated with tectonic processes, such as crustal strain and deep fluid migration.

Objective of short-term forecasting: The model is applied to enhance our capability for short-term and imminent earthquake forecasting (within a 45-day

window). So, the moving average window time is set to be shorter than the earthquake response time threshold (45 days).

Common practice in the field: The use of a moving average over this timescale (e.g., 14-day) is a well-established methodology in precursory fluid geochemical analysis, as evidenced by its application in numerous previous studies (Piersanti et al., 2016; Fu et al., 2017; Zhao et al., 2021). We use a 15-day window, which is more applicable to 3-day resolution data.

Operational utility for real-time monitoring: We specifically employed a backward-looking moving average because it is practically viable for real-time data monitoring and analysis. This approach allows for the continuous updating of the baseline trend as each new data point arrives, which is essential for timely earthquake forecasting.

**References:**

Fu, C., Yang, T., Tsai, M., Lee, L., Liu, T., Walia, V., Chen, C., Chang, W., Kumar, A., and Lai, T.: Exploring the relationship between soil degassing and seismic activity by continuous radon monitoring in the Longitudinal Valley of eastern Taiwan, Chemical Geology, 469, 163-175, https://doi.org/10.1016/j.chemgeo.2016.12.042, 2017.

Piersanti, A., Cannelli, V., and Galli, G.: The Pollino 2012 seismic sequence: clues from continuous radon monitoring, Solid Earth, 7(5), 1303-1316, https://doi.org/10.5194/se-7-1303-2016, 2016.

Zhao, Y., Liu, Z., Li, Y., Hu, L., Chen, Z., Sun, F., and Lu, C.: A case study of 10 years groundwater radon monitoring along the eastern margin of the Tibetan Plateau and in its adjacent regions: Implications for earthquake surveillance, Applied Geochemistry, 131, https://doi.org/10.1016/j.apgeochem.2021.105014, 2021.

8. *#Line 250 Please cite references here about this definition.*

**Response:** We included two new citations in the sentence on Line 250:

The cross-correlation function (Chatfield, 1975; Brockwell and Davis, 1991) is defined as:

**References:**

Brockwell, P. J., and Davis, R. A.: Time Series: Theory and Methods (Second Edition), Springer-Verlag, New York, 407pp, ISBN978-1-4419-0319-8, 1991.

Chatfield, C.: The Analysis of Time Series: Theory and Practice, Chapman and Hall, New York, 173pp, ISBN978-0-412-14180-5, 1975.

9. *#Line 315 This paragraph is more like results and discussion (limitation). It is not appropriate to present here.*

**Response:** We sincerely thank you for this insightful comment. We agree entirely that the paragraph in question, which discusses the limitations of our findings, is more appropriately placed in the Results and discussion section rather than where it was previously located. Our intention was not to present a full limitations section prematurely, but rather to use the inherent limitations of Bayesian analysis as a direct motivator and contrasting backdrop for introducing our detection model. However, we appreciate that deviating from standard structure can be disruptive. We have followed this suggestion and have moved this paragraph to the Results and discussion section of the manuscript.

10. *#Line 577 Please describe this conclusion in more detail.*

**Response:** We are grateful for your thoughtful comment. We agree completely that providing more detail will significantly strengthen the clarity and impact of our conclusion. In response, we have revised the conclusion in Line 577 to provide a more comprehensive and clearer summary:

The anomaly detection model demonstrates reliable real-time anomaly detection capabilities, with POD ranging from 0.83 to 0.95 and TS between 0.59 and 0.70, and it shows similar anomaly detection results across different springs to the same earthquake. The model identifies $Na^+$, $Ca^{2+}$, $Cl^-$, $SO_4^{2-}$, $\delta D$, and $\delta^{18}O$ can serve as effective indicators for strong earthquakes forecasting, all showing pre-earthquake high values and TS above 0.50. Among these, $\delta D$ and $\delta^{18}O$ exhibit higher sensitivity to seismic activity, characterized by multiple consecutive anomalies pre-earthquake.

---

## Author Comment (AC4)

**RESPONSES TO REVIEWER TWO'S COMMENTS**

We would like to express our sincere appreciation for your valuable comments and suggestions on our manuscript. We have carefully considered the comments and have revised the manuscript accordingly. The comments are laid out below in italicized font. Our response is given in normal font and changes/additions to the manuscript are given in the blue text.

**# Major comments:**

1. *#The introduction offers an overview of earthquake precursor research, referencing relevant studies (e.g., Chen, 2009; Pritchard et al., 2020). However, it lacks a comprehensive review of competing methodologies, such as Bayesian Change Point (BCP) analysis or geophysical approaches, which would better contextualize the proposed hydrochemical method. Additionally, while it notes challenges in isolating seismic precursors from complex fluid monitoring data, it fails to clearly articulate specific gaps in existing hydrochemical anomaly detection research. For example, it mentions limitations of single-indicator methods but does not quantify their shortcomings (e.g., false positive rates) or directly compare them to the multicomponent approach. This limited scope results in an underdeveloped problem statement, rendering the study's contribution ambiguous and weakening the justification for its methodological novelty.*

   **Response:** We thank the reviewer for this insightful and constructive feedback. We agree that a more comprehensive literature review and a clearer articulation of the research gap are essential to strengthen the introduction.

   The study of earthquake precursors relies significantly on various geophysical

approaches, including seismology, geomagnetism, geoelectrical methods, and ionospheric monitoring. Specifically, seismological methods focus on phenomena such as seismic activity analysis, b-value variations, and seismic wave velocity anomalies (Papadimitriou, 2008; Chen and Zhu, 2020). Geomagnetic techniques are dedicated to observing anomalies in the geomagnetic field (Chen et al., 2022), while geoelectric methods involve monitoring anomalies in ground resistivity and the geoelectric field (Sidorin, 2003; An et al., 2019). Ionospheric monitoring seeks information potentially related to earthquakes by analyzing parameters such as variations in Total Electron Content (TEC) (Zulhamidi et al., 2023; Nayak et al., 2024). These approaches are primarily effective for identifying long-term trend precursors and are thus more suitable for medium- to long-term earthquake forecasting. Nevertheless, for short-term and imminent earthquake forecasting, fluid geochemical methods exhibit unique advantages due to their sensitivity to changes in crustal stress.

To further enhance the analysis of data, Bayesian techniques are particularly valuable for addressing uncertainties, integrating multi-source data, and improving predictive accuracy (Zhang et al., 2016; Jiao and Shan, 2024). Specific applications include detecting ionospheric anomalies with Bayesian-LSTM models (Saqib et al., 2024), predicting GPS-TEC variations via Bayesian regularized backpropagation algorithms (Karatay and Gul, 2023), and analyzing groundwater geochemical composition using Bayesian mixture models (Chen et al., 2025), all of which contribute to more reliable identification of earthquake precursor signals.

In response to the comment on articulating research gaps, in the revised manuscript, we will explicitly state the shortcomings of single-index methods. Regarding the direct comparison with multi-component methods, to the best of our knowledge, there is a scarcity of published results applying such methods. This makes a direct, quantitative comparison challenging at this stage.

**References:**

Papadimitriou, P.: Identification of seismic precursors before large earthquakes:

Decelerating and accelerating seismic patterns. Journal of Geophysical Research: Solid Earth, 113(B4), https://doi.org/10.1029/2007jb005112, 2008.

An, Z., Zhan, Y., Fan, Y., Chen, Q., and Liu, J.: Investigation of the characteristics of geoelectric field earthquake precursors: a case study of the Pingliang observation station, China, Annals of Geophysics, 63(5), PA545, https://doi.org/10.4401/ag-7982, 2020.

Nayak, K., Romero-Andrade, R., Sharma, G., López-Urías, C., Trejo-Soto, M. E., and Vidal-Vega, A. I.: Evaluating Ionospheric Total Electron Content (TEC) Variations as Precursors to Seismic Activity: Insights from the 2024 Noto Peninsula and Nichinan Earthquakes of Japan, Atmosphere, 15(12), 1492, https://doi.org/10.3390/atmos15121492, 2024.

Sidorin, A. Ya.: Search for earthquake precursors in multidisciplinary data monitoring of geophysical and biological parameters, Natural Hazards and Earth System Sciences, 3(3/4), 153–158, https://doi.org/10.5194/nhess-3-153-2003, 2003.

Zulhamidi, N. F. I., Abdullah, M., Abdul Hamid, N. S., Yusof, K. A., and Bahari, S. A.: Investigating short-term earthquake precursors detection through monitoring of total electron content variation in ionosphere, Frontiers in Astronomy and Space Sciences, 10, https://doi.org/10.3389/fspas.2023.1166394, 2023.

Chen J, Zhu S.: Spatial and temporal b-value precursors preceding the 2008 Wenchuan, China, earthquake (Mw=7.9): implications for earthquake prediction, Geomatics, Natural Hazards and Risk, 11(1), 1196-1211, https://doi.org/10.1080/19475705.2020.1784297, 2020.

Chen, H., Han, P., Hattori, K.: Recent Advances and Challenges in the Seismo-Electromagnetic Study: A Brief Review, Remote Sensing, 14, 5893, https://doi.org/10.3390/rs14225893, 2022.

Chen, Y., Huang, F., Hu, L., Wang, Z., Yang, M., Hua, P., Sun, X., Zhu, S., Zhang, Y., Wu, X., Wang, Z., Xu, L., Han, K., Cui, B., Dong, H., Fei, B., and Zhou, Y.: Two Opposite Change Patterns Before Small Earthquakes Based on Consecutive Measurements of Hydrogen and Oxygen Isotopes at Two Seismic Monitoring

Sites in Northern Beijing, China. Geosciences, 15(6), https://doi.org/10.3390/geosciences15060192, 2025.

Jiao, Z., & Shan, X.: A Bayesian Approach for Forecasting the Probability of Large Earthquakes Using Thermal Anomalies from Satellite Observations, Remote Sensing, 16(9), https://doi.org/10.3390/rs16091542, 2024.

Karatay, S., & Gul, S. E.: Prediction of GPS-TEC on Mw > 5 Earthquake Days Using Bayesian Regularization Backpropagation Algorithm, IEEE Geoscience and Remote Sensing Letters, 20, 1-5, https://doi.org/10.1109/lgrs.2023.3262028, 2023.

Saqib, M., Şentürk, E., Arqim Adil, M., and Freeshah, M.: Seismo-ionospheric precursory detection using hybrid Bayesian-LSTM network model with uncertainty-boundaries and anomaly-intensity, Advances in Space Research, 74(4), 1828-1842, https://doi.org/10.1016/j.asr.2024.05.023, 2024.

Zhang, Y., Zhao, H., He, X., Pei, F.-D., & Li, G.-G.: Bayesian prediction of earthquake network based on space–time influence domain, Physica A: Statistical Mechanics and its Applications, 445, 138-149, https://doi.org/10.1016/j.physa.2015.11.006, 2016.

2. *#The earthquake selection method employs Dobrovolsky's preparation zone radius formula, assuming isotropic subsurface structures. This oversimplification disregards the anisotropic complexities of faults and aquifers at the Xiaojiang-Red River Fault (XJF-RRF) intersection, potentially leading to inaccurate event selection. The manuscript neither justifies this assumption nor evaluates its sensitivity, which compromises the reliability of correlations between hydrochemical anomalies and seismic events.*

**Response:** Thank you for your insightful comment regarding the use of Dobrovolsky's formula for estimating the earthquake preparation zone radius. We agree that the assumption of isotropic subsurface structures is a simplification, particularly in complex fault-aquifer systems such as the Xiaojiang-Red River Fault (XJF-RRF)

intersection, where anomalies may be more likely to occur along the strike of fault.

In our study, we adopted Dobrovolsky's empirical formula based on its theoretical foundation and widespread validation in the literature. As derived in Dobrovolsky et al. (1979), the formula originates from a mechanical model that treats the preparation zone as a "soft inclusion" within an elastic half-space. The calculation corresponds to a special case of a homogeneous isotropic inclusion where only the shear modulus decreases, under the action of shear stresses applied at infinity. This formulation allows estimation of surface deformations and tilts as a function of both earthquake magnitude and epicentral distance. Crucially, it was shown that the precursors of other physical nature fall into this circle. As emphasized in the original work, the primary purpose of the model is not to capture intricate anisotropic fault structures, but rather to estimate the "zone of effective manifestation of the precursor deformations"—termed the "strain radius." Thus, even in anisotropic media, the formula provides a robust reference scale for the maximum circular area potentially influenced by the source process.

In long-term studies of seismic fluid geochemistry, comprehensive comparisons of empirical formulas used in precursor research have identified Dobrovolsky's formula as the theoretical foundation for numerous geochemical earthquake prediction methods, demonstrating relatively broad applicability (Li et al., 2023). This foundation has been applied in forecasting studies across various seismic regions worldwide, such as those focusing on radon emanation and hydrochemistry (Hashemi et al., 2012; Fu et al., 2017; Barkat et al., 2018; Süer et al., 2020; Zhang et al., 2020; Zhao et al., 2021; Zhou et al., 2021; Seminsky and Seminsky, 2024; Zhu et al., 2024). Based on long-term observations of 27 radon-involved earthquake cases in China between 1997 and 2020, 9 widely applied prediction methods were evaluated. The results showed that Dobrovolsky's formula achieved the highest applicability rate, reaching 96.30% (Li et al., 2023). A recent study on $M$w 5.0 earthquakes in Western Türkiye (Yakupoglu et al., 2025) demonstrated that pre-earthquake hydrogeochemical anomalies were detected within the strain radius predicted by Dobrovolsky's formula but well beyond the smaller radius estimated by the geologically-constrained model ($D = 10^{0.28M+0.25}$) by Martinelli

and Tamburello (2020). This finding, observed in a complex tectonic setting, supports the validity and practical utility of Dobrovolsky's formula for defining the potential extent of preparatory processes.

We acknowledge that the isotropic assumption may not fully capture the anisotropic nature of fault zones and aquifer systems. Although Dobrovolsky's formula has been extensively validated by global observations, we have explicitly addressed this limitation in the Limitations section of our manuscript. Based on our results, we note that some false positive earthquake events may be attributed to the isotropic assumption of the earthquake selection criteria, which cause earthquakes located beyond the calculated radius along the same fault zone to be overlooked. As described on line 553: "Approximately 30 days after the last multicomponent synchronised false alarms at the two thermal springs, an $M$ 4.1 earthquake occurred, with an epicenter located outside the radius of the earthquake preparation zone." Therefore, we have emphasized that future work should incorporate more complex, anisotropic models to improve accuracy in active tectonic regions.

Based on the above statement, we have added more details on Lines 176-183:

To identify earthquakes potentially influencing hydrochemical component variations and to establish a precise correlation between hydrochemical changes and seismic activity, a screening method based on the preparation zone radius formula (Dobrovolsky et al., 1979) was employed:

$$R = 10^{0.43M} \tag{2}$$

where $M$ represents the earthquake magnitude, and $R$ denotes the radius (in km) of the earthquake preparation zone. The method provides an empirically validated and widely adopted reference scale for selecting potentially correlated earthquake events (Li et al., 2023; Zhu et al., 2024; Yakupoglu et al., 2025).

On Lines 555-557:

Current earthquake screening method assumes an isotropic underground structure although effective and widely used. Therefore, the algorithm requires optimisation based on the specific geological background in further research.

**References:**

Barkat, A., Ali, A., Hayat, U., Crowley, Q. G., Rehman, K., Siddique, N., Haidar, T., and Iqbal, T.: Time series analysis of soil radon in Northern Pakistan: Implications for earthquake forecasting, Applied Geochemistry, 97, 197-208, https://doi.org/10.1016/j.apgeochem.2018.08.016, 2018.

Fu, C., Yang, T., Tsai, M., Lee, L., Liu, T., Walia, V., Chen, C., Chang, W., Kumar, A., and Lai, T.: Exploring the relationship between soil degassing and seismic activity by continuous radon monitoring in the Longitudinal Valley of eastern Taiwan, Chemical Geology, 469, 163-175, https://doi.org/10.1016/j.chemgeo.2016.12.042, 2017.

Hashemi, S. M., Negarestani, A., Namvaran, M., and Musavi Nasab, S. M.: An analytical algorithm for designing radon monitoring network to predict the location and magnitude of earthquakes, Journal of Radioanalytical and Nuclear Chemistry, 295(3), 2249-2262, https://doi.org/10.1007/s10967-012-2310-0, 2012.

Li Y., Fang Z., Zhang C., Li J., Bao Z., Zhang X., Liu Z., Zhou X., Chen Z., and Du J.: Research progress and prospect of seismic fluid geochemistry in short-imminent earthquake prediction (in Chinese with English abstract), Seismology and Geology, 45(3), 593-621, https://doi:10.3969 /j.issn.0253- 4967.2023.03.001, 2023.

Martinelli, G., and Tamburello, G.: Geological and Geophysical Factors Constraining the Occurrence of Earthquake Precursors in Geofluids: A Review and Reinterpretation, Frontiers in Earth Science, 8, https://doi.org/10.3389/feart.2020.596050, 2020.

Seminsky, A. K., Seminsky, K. Z.: Time-dependent variations of groundwater radon: Insights from a twelve-year study in the Baikal region, East Siberia, Russia. J Environ Radioact, 278, 107509, https://doi.org/10.1016/j.jenvrad.2024.107509, 2024.

Süer, S., Wiersberg, T., Güleç, N., Erzinger, J., and Parlaktuna, M.: Real-time gas monitoring at the Tekke Hamam geothermal field (Western Anatolia, Turkey): an

assessment in relation to local seismicity, Natural Hazards, 104(2), 1655-1678, https://doi.org/10.1007/s11069-020-04238-8, 2020.

Yakupoğlu, N., Sabuncu, A., Erbil, C., Kırkan E., Çetin H., and İnan S.: Pre-earthquake hydrogeochemical anomalies in spring waters: two distinctive cases from western Türkiye, Journal of Hydrology, 662, https://doi.org/10.1016/j.jhydrol.2025.133920, 2025.

Zhang, L., Guo, L., Wang, Y., Liu, D., Liu, Y., and Li, J.: Continuous monitoring of hydrogen and oxygen stable isotopes in a hot spring: Significance for distant earthquakes, Applied Geochemistry, 112, https://doi.org/10.1016/j.apgeochem.2019.104488, 2020.

Zhao, Y., Liu, Z., Li, Y., Hu, L., Chen, Z., Sun, F., and Lu, C.: A case study of 10 years groundwater radon monitoring along the eastern margin of the Tibetan Plateau and in its adjacent regions: Implications for earthquake surveillance, Applied Geochemistry, 131, https://doi.org/10.1016/j.apgeochem.2021.105014, 2021.

Zhou, Z., Zhong, J., Zhao, J., Yan, R., Tian, L., and Fu, H.: Two Mechanisms of Earthquake-Induced Hydrochemical Variations in an Observation Well, water, 13(17), https://doi.org/10.3390/w13172385, 2021.

Zhu, R., Yang, F., Zhou, X., Tian, J., Zhang, Y., He, M., Li, J., Dong, J., and Li, Y.: Anomaly Detection Using Machine Learning in Hydrochemical Data From Hot Springs: Implications for Earthquake Prediction, Water Resources Research, 60(6), https://doi.org/10.1029/2023wr034748, 2024.

3. *#The 15-day moving average effectively reduces rainfall-induced noise, as evidenced by low cross-correlation coefficients ($\pm 0.2$). However, the method lacks validation against other environmental factors, such as temperature or barometric pressure, or long-term trends that may influence hydrochemical signals. Without a control dataset or comparison with alternative filtering techniques (e.g., wavelet transforms), confidence in the denoising process is limited, undermining the*

*robustness of the data preprocessing methodology.*

**Response:** We sincerely thank the reviewer for this insightful comment and for acknowledging the effectiveness of the 15-day moving average. We agree that validating the denoising process is crucial for robustness. We have carefully considered your points regarding the potential influences of environmental factors beyond rainfall, as well as the validation of the denoising approach. We have now performed additional analyses and revised the manuscript accordingly to address these concerns point-by-point.

We would like to clarify that the hydrochemical component of the thermal spring water in our study area is primarily controlled by deep geothermal processes and lithology of the surrounding rocks. To further substantiate this, we incorporated concurrent meteorological data. Specifically, comparative plots (Figure S6) between hydrochemical data (taking $Na^+$ time series from Qujiang spring as an example) and local meteorological data (including temperature and atmospheric pressure) consistently show weak correlation, confirming that these factors do not significantly influence the variability in hydrochemical component concentrations. Moreover, partial decreases in $Na^+$ concentration following rainfall events are observable. These confirm our focus on rainfall as the dominant environmental noise source.

[Figure]

Figure S6. Time series of $Na^+$, alongside corresponding rainfall (R), temperature (T), and

atmospheric pressure (P) for Qujiang spring.

While the 15-day moving average effectively reduces high-frequency noise related to rainfall pulses, we agree that additional validation strengthens methodological robustness. Accordingly, we have further applied both Fast Fourier Transform (FFT) low-pass filtering and a 3-level wavelet denoising technique (using DB5 wavelet with 20% threshold) to the same dataset (Figure S7, again using the $Na^+$ series from Qujiang as an example). The results demonstrate approximately consistent signal smoothing across all three methods, thereby reinforcing the reliability of the moving average approach for denoising process. Although the backward moving average introduces a slight phase lag, this method is better suited to the real-time anomaly detection framework of our detection model.

[Figure]

Figure S7. Comparison of denoising results using 15-day moving average, Fast Fourier Transform, and 3-level Wavelet.

We have revised the manuscript on line 225 to include a brief comparison of these filtering techniques and to more explicitly state the rationale for focusing on rainfall-induced noise:

The thermal spring water in the study area originates from atmospheric precipitation recharge. It circulates deeply through faults, is heated by geothermal energy, and then discharges at the surface, with its hydrochemical composition mainly

determined by the lithology of the surrounding rocks (Shao et al., 2024). Consequently, ambient temperature and atmospheric pressure at the spring outlet have a negligible effect on the hydrochemistry. However, rainfall serves not only as the primary water source but also accelerates groundwater circulation, promotes shallow infiltration, and mixes with thermal waters (Taylor et al., 2012; Hosono et al., 2020; Colman et al., 2021). This process can potentially obscure deep-seated earthquake preparatory signals carried by the thermal spring. Consequently, this study focuses on assessing the potential perturbations induced by rainfall on thermal spring hydrochemistry. As shown in the comparative analysis of hydrochemical and meteorological data (Figure S6), rainfall is the dominant interfering factor, with ion concentrations partially decreasing following events, whereas the effects of temperature and pressure are negligible. Unlike temperature and pressure, rainfall causes pulsed disturbances, typically manifesting as intermittent spikes followed by extended zero-value intervals in sampling data. To suppress high-frequency noise from short-term environmental disturbances such as rainfall while preserving mid- to low-frequency tectonic signals, a 15-day backward moving average is applied to process the 3-day resolution hydrochemistry data. This method is better suited to the real-time anomaly detection framework of detection model. To validate the robustness of the denoising process, the results obtained from the moving average were compared with those derived from Fast Fourier Transform low-pass filtering and wavelet-based denoising techniques (Figure S7). The approximately consistent outcomes across all methods confirm the suitability of the

moving average approach for suppressing high-frequency noise.

$$MA(t) = \frac{1}{15} \sum_{t-14}^{t} Dr(t) \tag{5}$$

where *MA* is the 15-day moving average, and *Dr* is the daily raw data.

4. *#The optimization of parameters p1 and p3 is thoroughly described but lacks transparency regarding the selection of parameter ranges. The manuscript does not explore alternative optimization methods, such as grid search with cross-validation, nor does it assess the sensitivity of results to parameter variations beyond the TS presented in Figure 6. This omission raises concerns about the robustness and reproducibility of the anomaly detection model. Furthermore, using the entire dataset for parameter optimization introduces a significant risk of overfitting, which is not addressed, further undermining the model's reliability.*

**Response:** We thank the reviewer for their thorough review and valuable comments, which have helped us significantly improve the manuscript.

1) We acknowledge that the anomaly detection model is an application-specific model for the seismic industry, not a general-purpose computer model, with the goal of exploring relatively effective methods for earthquake forecasting. The parameter ranges were not chosen arbitrarily but were based on certain seismological rationale and operational framework of model. Please allow us to clarify the rationale for each parameter:

Parameters p1 and p3 represent multiples of the sliding window values. The model logic requires a baseline multiplier exceeding 1.0 to define a meaningful deviation from threshold. Values below 1.0 would inappropriately trigger alarms for values falling below the average, which does not represent the target earthquake anomaly. The upper bound of 1.2 was determined empirically during the optimization process, where it was consistently observed that model performance degraded beyond this value, thereby

establishing a natural performance-based limit for these threshold multipliers.

Parameter p2 defines the anomaly duration interval, which must not exceed the seismic response time threshold of 45, otherwise it becomes meaningless.

Parameter p4 represent post-earthquake sensitivity adjustment multiplier. The purpose of p4 is to increase the thresholds (p1 and p3) following an earthquake to reduce sensitivity to post-earthquake disturbances. Therefore, a value greater than 1.0 is physically necessary to achieve this effect. However, an excessively high value for p4 would oversuppress the model's sensitivity, potentially causing it to miss genuine post-earthquake anomalies and thus defeating its intended purpose.

Parameter p5 represent magnitude-scaling factor for post-earthquake duration. Its value must exceed 1.0 to ensure the period ($p5^{\wedge}M$) scales appropriately with increasing earthquake magnitude (*M*). An appropriate upper bound is applied to prevent unrealistically long periods that would lack physical justification.

In summary, the lower bounds for parameters are defined by the model's foundational logic, while the upper bounds are determined by actual model performance test results and fundamental constraints of seismology.

We have briefly supplemented the basis for the selection of parameter ranges on line 427:

The parameter ranges were based on certain seismological rationale, operational framework of model, and actual model performance test results. For optimisation involving ion concentration data, the model applies parameter values ranging from 1.00 to 1.20 in steps of 0.01.

2) The reviewer rightly pointed out that the original manuscript did not assess the sensitivity of the results to parameter variations beyond the Threat Score (TS) presented in Figure 6. We have conducted additional sensitivity analyses by testing the model's performance under variations of two key parameters. The results are provided in two new supplementary tables (Tables S3 and S4), which present the corresponding values for correct alarms (NA), false alarms (NB), and missed alarms (NC), False alarm rate (FAR), Missed alarm rate (MAR), Probability of detection (POD), and TS values under

different parameter settings. The best values are denoted by bold.

Table S3. Model performance metrics (NA, NB, NC, FAR, MAR, POD, TS) under varying parameters p1 and p3 for Na$^+$ at Qujiang Spring.

| p1 | p3 | NB | NA | NA+NC | FAR | MAR | POD | TS |
|------|------|----|----|-------|------|------|------|------|
| 1 | 1 | 19 | 20 | 22 | 0.49 | **0.09** | **0.91** | 0.49 |
| 1 | 1.01 | 12 | 18 | 22 | 0.40 | 0.18 | 0.82 | 0.53 |
| 1 | 1.02 | 6 | 14 | 22 | 0.30 | 0.36 | 0.64 | 0.50 |
| 1 | 1.03 | 5 | 11 | 22 | 0.31 | 0.50 | 0.50 | 0.41 |
| 1 | 1.04 | 4 | 9 | 22 | 0.31 | 0.59 | 0.41 | 0.35 |
| 1 | 1.05 | 3 | 7 | 22 | 0.30 | 0.68 | 0.32 | 0.28 |
| 1 | 1.06 | 2 | 5 | 22 | 0.29 | 0.77 | 0.23 | 0.21 |
| 1 | 1.07 | 1 | 4 | 22 | 0.20 | 0.82 | 0.18 | 0.17 |
| 1 | 1.08 | 1 | 4 | 22 | 0.20 | 0.82 | 0.18 | 0.17 |
| 1.01 | 1 | 14 | 19 | 22 | 0.42 | 0.14 | 0.86 | 0.53 |
| 1.01 | 1.01 | 8 | 16 | 22 | 0.33 | 0.27 | 0.73 | 0.53 |
| 1.01 | 1.02 | 5 | 12 | 22 | 0.29 | 0.45 | 0.55 | 0.44 |
| 1.01 | 1.03 | 5 | 5 | 22 | 0.50 | 0.77 | 0.23 | 0.19 |
| 1.01 | 1.04 | 4 | 6 | 22 | 0.40 | 0.73 | 0.27 | 0.23 |
| 1.01 | 1.05 | 1 | 6 | 22 | **0.14** | 0.73 | 0.27 | 0.26 |
| 1.01 | 1.06 | 1 | 5 | 22 | 0.17 | 0.77 | 0.23 | 0.22 |
| 1.01 | 1.07 | 1 | 4 | 22 | 0.20 | 0.82 | 0.18 | 0.17 |
| 1.01 | 1.08 | 1 | 2 | 22 | 0.33 | 0.91 | 0.09 | 0.09 |
| 1.02 | 1 | 8 | 16 | 22 | 0.33 | 0.27 | 0.73 | 0.53 |
| 1.02 | 1.01 | 4 | 14 | 22 | 0.22 | 0.36 | 0.64 | **0.54** |
| 1.02 | 1.02 | 4 | 11 | 22 | 0.27 | 0.50 | 0.50 | 0.42 |
| 1.02 | 1.03 | 4 | 7 | 22 | 0.36 | 0.68 | 0.32 | 0.27 |
| 1.02 | 1.04 | 2 | 6 | 22 | 0.25 | 0.73 | 0.27 | 0.25 |
| 1.02 | 1.05 | 1 | 5 | 22 | 0.17 | 0.77 | 0.23 | 0.22 |
| 1.02 | 1.06 | 1 | 5 | 22 | 0.17 | 0.77 | 0.23 | 0.22 |
| 1.02 | 1.07 | 1 | 4 | 22 | 0.20 | 0.82 | 0.18 | 0.17 |
| 1.02 | 1.08 | 1 | 2 | 22 | 0.33 | 0.91 | 0.09 | 0.09 |
| 1.03 | 1 | 4 | 13 | 22 | 0.24 | 0.41 | 0.59 | 0.50 |
| 1.03 | 1.01 | 4 | 12 | 22 | 0.25 | 0.45 | 0.55 | 0.46 |
| 1.03 | 1.02 | 2 | 10 | 22 | 0.17 | 0.55 | 0.45 | 0.42 |
| 1.03 | 1.03 | 2 | 6 | 22 | 0.25 | 0.73 | 0.27 | 0.25 |
| 1.03 | 1.04 | 2 | 6 | 22 | 0.25 | 0.73 | 0.27 | 0.25 |
| 1.03 | 1.05 | 1 | 5 | 22 | 0.17 | 0.77 | 0.23 | 0.22 |
| 1.03 | 1.06 | 1 | 5 | 22 | 0.17 | 0.77 | 0.23 | 0.22 |
| 1.03 | 1.07 | 1 | 3 | 22 | 0.25 | 0.86 | 0.14 | 0.13 |
| 1.03 | 1.08 | 1 | 2 | 22 | 0.33 | 0.91 | 0.09 | 0.09 |

| | | | | | | | |
|------|------|---|----|----|------|------|------|------|
| 1.04 | 1    | 2 | 10 | 22 | 0.17 | 0.55 | 0.45 | 0.42 |
| 1.04 | 1.01 | 2 | 8  | 22 | 0.20 | 0.64 | 0.36 | 0.33 |
| 1.04 | 1.02 | 2 | 7  | 22 | 0.22 | 0.68 | 0.32 | 0.29 |
| 1.04 | 1.03 | 2 | 6  | 22 | 0.25 | 0.73 | 0.27 | 0.25 |
| 1.04 | 1.04 | 2 | 5  | 22 | 0.29 | 0.77 | 0.23 | 0.21 |
| 1.04 | 1.05 | 1 | 5  | 22 | 0.17 | 0.77 | 0.23 | 0.22 |
| 1.04 | 1.06 | 1 | 5  | 22 | 0.17 | 0.77 | 0.23 | 0.22 |
| 1.04 | 1.07 | 1 | 3  | 22 | 0.25 | 0.86 | 0.14 | 0.13 |
| 1.04 | 1.08 | 1 | 2  | 22 | 0.33 | 0.91 | 0.09 | 0.09 |
| 1.05 | 1    | 2 | 6  | 22 | 0.25 | 0.73 | 0.27 | 0.25 |
| 1.05 | 1.01 | 2 | 6  | 22 | 0.25 | 0.73 | 0.27 | 0.25 |
| 1.05 | 1.02 | 2 | 6  | 22 | 0.25 | 0.73 | 0.27 | 0.25 |
| 1.05 | 1.03 | 2 | 6  | 22 | 0.25 | 0.73 | 0.27 | 0.25 |
| 1.05 | 1.04 | 2 | 5  | 22 | 0.29 | 0.77 | 0.23 | 0.21 |
| 1.05 | 1.05 | 1 | 5  | 22 | 0.17 | 0.77 | 0.23 | 0.22 |
| 1.05 | 1.06 | 1 | 5  | 22 | 0.17 | 0.77 | 0.23 | 0.22 |
| 1.05 | 1.07 | 1 | 3  | 22 | 0.25 | 0.86 | 0.14 | 0.13 |
| 1.05 | 1.08 | 1 | 2  | 22 | 0.33 | 0.91 | 0.09 | 0.09 |
| 1.06 | 1    | 2 | 5  | 22 | 0.29 | 0.77 | 0.23 | 0.21 |
| 1.06 | 1.01 | 2 | 5  | 22 | 0.29 | 0.77 | 0.23 | 0.21 |
| 1.06 | 1.02 | 2 | 5  | 22 | 0.29 | 0.77 | 0.23 | 0.21 |
| 1.06 | 1.03 | 2 | 5  | 22 | 0.29 | 0.77 | 0.23 | 0.21 |
| 1.06 | 1.04 | 2 | 4  | 22 | 0.33 | 0.82 | 0.18 | 0.17 |
| 1.06 | 1.05 | 1 | 4  | 22 | 0.20 | 0.82 | 0.18 | 0.17 |
| 1.06 | 1.06 | 1 | 4  | 22 | 0.20 | 0.82 | 0.18 | 0.17 |
| 1.06 | 1.07 | 1 | 3  | 22 | 0.25 | 0.86 | 0.14 | 0.13 |
| 1.06 | 1.08 | 1 | 2  | 22 | 0.33 | 0.91 | 0.09 | 0.09 |

Table S4. Model performance metrics (NA, NB, NC, FAR, MAR, POD, TS) under varying parameters p1 and p3 for $SO_4^{2-}$ at Wana Spring.

| p1 | p3   | NB | NA | NA+NC | FAR  | MAR      | POD      | TS   |
|----|------|----|----|-------|------|----------|----------|------|
| 1  | 1    | 12 | 10 | 12    | 0.55 | **0.17** | **0.83** | 0.42 |
| 1  | 1.01 | 11 | 10 | 12    | 0.52 | **0.17** | **0.83** | 0.43 |
| 1  | 1.02 | 11 | 10 | 12    | 0.52 | **0.17** | **0.83** | 0.43 |
| 1  | 1.03 | 11 | 10 | 12    | 0.52 | **0.17** | **0.83** | 0.43 |
| 1  | 1.04 | 9  | 8  | 12    | 0.53 | 0.33     | 0.67     | 0.38 |
| 1  | 1.05 | 8  | 8  | 12    | 0.50 | 0.33     | 0.67     | 0.40 |
| 1  | 1.06 | 7  | 8  | 12    | 0.47 | 0.33     | 0.67     | 0.42 |
| 1  | 1.07 | 7  | 6  | 12    | 0.54 | 0.50     | 0.50     | 0.32 |
| 1  | 1.08 | 6  | 6  | 12    | 0.50 | 0.50     | 0.50     | 0.33 |

| 1.01 | 1 | 11 | 10 | 12 | 0.52 | **0.17** | **0.83** | 0.43 |
|------|------|----|----|----|------|------|------|------|
| 1.01 | 1.01 | 11 | 10 | 12 | 0.52 | **0.17** | **0.83** | 0.43 |
| 1.01 | 1.02 | 10 | 10 | 12 | 0.50 | **0.17** | **0.83** | 0.45 |
| 1.01 | 1.03 | 10 | 10 | 12 | 0.50 | **0.17** | **0.83** | 0.45 |
| 1.01 | 1.04 | 8 | 8 | 12 | 0.50 | 0.33 | 0.67 | 0.40 |
| 1.01 | 1.05 | 8 | 8 | 12 | 0.50 | 0.33 | 0.67 | 0.40 |
| 1.01 | 1.06 | 7 | 8 | 12 | 0.47 | 0.33 | 0.67 | 0.42 |
| 1.01 | 1.07 | 7 | 6 | 12 | 0.54 | 0.50 | 0.50 | 0.32 |
| 1.01 | 1.08 | 6 | 6 | 12 | 0.50 | 0.50 | 0.50 | 0.33 |
| 1.02 | 1 | 10 | 10 | 12 | 0.50 | **0.17** | **0.83** | 0.45 |
| 1.02 | 1.01 | 10 | 10 | 12 | 0.50 | **0.17** | **0.83** | 0.45 |
| 1.02 | 1.02 | 10 | 10 | 12 | 0.50 | **0.17** | **0.83** | 0.45 |
| 1.02 | 1.03 | 10 | 10 | 12 | 0.50 | **0.17** | **0.83** | 0.45 |
| 1.02 | 1.04 | 8 | 8 | 12 | 0.50 | 0.33 | 0.67 | 0.40 |
| 1.02 | 1.05 | 8 | 8 | 12 | 0.50 | 0.33 | 0.67 | 0.40 |
| 1.02 | 1.06 | 7 | 8 | 12 | 0.47 | 0.33 | 0.67 | 0.42 |
| 1.02 | 1.07 | 7 | 6 | 12 | 0.54 | 0.50 | 0.50 | 0.32 |
| 1.02 | 1.08 | 6 | 6 | 12 | 0.50 | 0.50 | 0.50 | 0.33 |
| 1.03 | 1 | 9 | 10 | 12 | 0.47 | **0.17** | **0.83** | 0.48 |
| 1.03 | 1.01 | 8 | 10 | 12 | 0.44 | **0.17** | **0.83** | 0.50 |
| 1.03 | 1.02 | 7 | 9 | 12 | 0.44 | 0.25 | 0.75 | 0.47 |
| 1.03 | 1.03 | 7 | 9 | 12 | 0.44 | 0.25 | 0.75 | 0.47 |
| 1.03 | 1.04 | 7 | 8 | 12 | 0.47 | 0.33 | 0.67 | 0.42 |
| 1.03 | 1.05 | 7 | 8 | 12 | 0.47 | 0.33 | 0.67 | 0.42 |
| 1.03 | 1.06 | 6 | 8 | 12 | 0.43 | 0.33 | 0.67 | 0.44 |
| 1.03 | 1.07 | 6 | 6 | 12 | 0.50 | 0.50 | 0.50 | 0.33 |
| 1.03 | 1.08 | 5 | 6 | 12 | 0.45 | 0.50 | 0.50 | 0.35 |
| 1.04 | 1 | 9 | 10 | 12 | 0.47 | **0.17** | **0.83** | 0.48 |
| 1.04 | 1.01 | 7 | 10 | 12 | **0.41** | **0.17** | **0.83** | **0.53** |
| 1.04 | 1.02 | 7 | 9 | 12 | 0.44 | 0.25 | 0.75 | 0.47 |
| 1.04 | 1.03 | 7 | 9 | 12 | 0.44 | 0.25 | 0.75 | 0.47 |
| 1.04 | 1.04 | 7 | 8 | 12 | 0.47 | 0.33 | 0.67 | 0.42 |
| 1.04 | 1.05 | 7 | 8 | 12 | 0.47 | 0.33 | 0.67 | 0.42 |
| 1.04 | 1.06 | 6 | 8 | 12 | 0.43 | 0.33 | 0.67 | 0.44 |
| 1.04 | 1.07 | 6 | 6 | 12 | 0.50 | 0.50 | 0.50 | 0.33 |
| 1.04 | 1.08 | 5 | 6 | 12 | 0.45 | 0.50 | 0.50 | 0.35 |
| 1.05 | 1 | 9 | 9 | 12 | 0.50 | 0.25 | 0.75 | 0.43 |
| 1.05 | 1.01 | 8 | 9 | 12 | 0.47 | 0.25 | 0.75 | 0.45 |
| 1.05 | 1.02 | 8 | 9 | 12 | 0.47 | 0.25 | 0.75 | 0.45 |
| 1.05 | 1.03 | 7 | 9 | 12 | 0.44 | 0.25 | 0.75 | 0.47 |
| 1.05 | 1.04 | 7 | 8 | 12 | 0.47 | 0.33 | 0.67 | 0.42 |
| 1.05 | 1.05 | 7 | 8 | 12 | 0.47 | 0.33 | 0.67 | 0.42 |
| 1.05 | 1.06 | 6 | 8 | 12 | 0.43 | 0.33 | 0.67 | 0.44 |

| | | | | | | | | |
|---|---|---|---|---|---|---|---|---|
| 1.05 | 1.07 | 6 | 6 | 12 | 0.50 | 0.50 | 0.50 | 0.33 |
| 1.05 | 1.08 | 5 | 6 | 12 | 0.45 | 0.50 | 0.50 | 0.35 |
| 1.06 | 1 | 7 | 8 | 12 | 0.47 | 0.33 | 0.67 | 0.42 |
| 1.06 | 1.01 | 7 | 8 | 12 | 0.47 | 0.33 | 0.67 | 0.42 |
| 1.06 | 1.02 | 6 | 7 | 12 | 0.46 | 0.42 | 0.58 | 0.39 |
| 1.06 | 1.03 | 6 | 7 | 12 | 0.46 | 0.42 | 0.58 | 0.39 |
| 1.06 | 1.04 | 6 | 6 | 12 | 0.50 | 0.50 | 0.50 | 0.33 |
| 1.06 | 1.05 | 6 | 6 | 12 | 0.50 | 0.50 | 0.50 | 0.33 |
| 1.06 | 1.06 | 5 | 6 | 12 | 0.45 | 0.50 | 0.50 | 0.35 |
| 1.06 | 1.07 | 5 | 5 | 12 | 0.50 | 0.58 | 0.42 | 0.29 |
| 1.06 | 1.08 | 4 | 5 | 12 | 0.44 | 0.58 | 0.42 | 0.31 |

The supplementary results confirm that the Threat Score (TS) provides the most robust evaluation of model performance. We selected the Threat Score (TS=NA/(NA+NB+NC)) as the primary metric because it provides a comprehensive evaluation by integrating NA, NB, and NC, making it particularly suitable for evaluating model performance on imbalanced data. It is important to note that the parameter sensitivity tables only present results for individual components to clearly illustrate the parameter selection process for each component. The multi-component joint model results tend to perform significantly better than any single-component results, since not every component achieves ideal anomaly detection result on its own.

3) We sincerely appreciate the reviewer's concern regarding overfitting, which is indeed a critical issue in any data-driven modeling study. The decision to use the entire dataset was primarily driven by its limited size, a common challenge in earthquake monitoring applications. Acquiring additional data is often prohibitively expensive or operationally infeasible, particularly for rare target events such as earthquakes. Specifically, for one hydrochemical component analyzed over a five-year monitoring period, the dataset consists of only 604 samples, which include just 22 potential earthquake events. Under such constraints, partitioning the data into a hold-out test set could easily result in a highly unrepresentative subset that might contain 0 earthquake events, thereby rendering performance evaluation unreliable or meaningless. Using the entire dataset enables a more stable and meaningful evaluation of model performance.

To mitigate the risk of overfitting, we developed a parsimonious model with a

limited number of parameters, in accordance with industry constraints. Furthermore, the model framework we adopted was designed to incorporate seismological interpretability and empirical predictive experience where feasible, thereby enhancing practical applicability and reducing the model's propensity to overfit compared to more complex models.

5. *#The study relies on data from only two thermal springs and a limited dataset (22 and 12 M ≥ 4 earthquakes, respectively), restricting the generalizability of findings to other tectonic settings or regional springs. The manuscript does not address how site-specific factors, such as lithology or fault geometry, might limit the model's applicability, thus diminishing its broader scientific impact. Furthermore, the abstract claims that $Na^+$, $Ca^{2+}$, $Cl^-$, $SO_4^{2-}$, $\delta D$, and $\delta^{18}O$ are sensitive indicators for earthquake forecasting, but this assertion is likely valid only for the studied springs. Additionally, the shorter time series for $\delta D$ and $\delta^{18}O$ in Figure 2 undermines their reliability as sensitive indicators.*

   **Response:** Thank you for your thoughtful comments regarding the generalizability and site-specific nature of our study. We would like to clarify that the two thermal springs were not selected arbitrarily. Their selection was based on a comprehensive survey of thermal springs within the study area, followed by a thorough analysis of their hydrogeochemical characteristics and tectonic activity (Shao et al., 2024). As described in the Geological setting section, these springs were chosen as long-term monitoring sites precisely because of their representative hydrological and tectonic conditions, which make them suitable for investigating earthquake-related hydrochemical changes. So, we agree that the identified sensitive indicators ($Na^+$, $Ca^{2+}$, $Cl^-$, $SO_4^{2-}$, $\delta D$, and $\delta^{18}O$) are likely most applicable to the specific thermal springs or region studied. We have revised the manuscript to ensure this is clearly stated and to avoid overgeneralization.

   However, the core value of our model lies not in universally transferring these

specific indicators, but in providing a methodological framework. For other regions, researchers can apply this model to their own continuous monitoring data from local springs. By doing so, they can identify which parameters are sensitive within their specific geological and hydrogeological context, thereby addressing differences in geological settings.

We completely agree that the number of earthquake events in our dataset is limited. This is indeed a common challenge in the field of earthquake hydrogeochemistry, given the inherent difficulty in obtaining continuous hydrogeological data, the low probability and uncertainty of earthquake occurrence. For example, even over a 14-year monitoring period in a seismically active region like Iceland, only about 5 $M \geq 5$ earthquakes might be recorded (Skelton et al., 2024). Similarly, at the junction of the Tibetan Plateau and the Yunnan-Guizhou Plateau, just 7 events of $M \geq 5$ were recorded over an 11-year period (Feng et al., 2022). In this context, our continuous observational dataset is relatively long-term and detailed for the earthquake hydrogeochemistry field, yet we acknowledge that data scarcity is a fundamental constraint in earthquake industry.

The reason for selecting geologically complex areas for this study is twofold. Firstly, these regions often have intricate seismogenic mechanisms, making it difficult to distinguish the timing, direction, and amplitude of anomalies based on physical mechanisms alone. Short-term, mechanism-based pre-earthquake anomaly analysis is exceptionally challenging in such settings. Secondly, precisely because these tectonically complex areas experience frequent earthquakes, they provide a relatively larger number of earthquake events within a monitoring period, offering a richer sample set for developing models. And there is a practical need for monitoring and risk mitigation in these seismically active regions. Therefore, the original intention of our model was not to rely on a fully understood physical mechanism and geological conditions first, but to explore a feasible method for extracting potential anomalies in scenarios where the underlying mechanisms are still unclear.

Based on the above statement, we have revised the manuscript on line 26:

The model identifies $Na^+$, $Ca^{2+}$, $Cl^-$, $SO_4^{2-}$, $\delta D$, and $\delta^{18}O$ as sensitive indicators

for strong earthquake forecasting in the study area.

On line 577:

The anomaly detection model demonstrates reliable real-time anomaly detection capabilities and identifies $Na^+$, $Ca^{2+}$, $Cl^-$, $SO_4^{2-}$, $\delta D$, and $\delta^{18}O$ as effective indicators for strong earthquake forecasting in the study area, with $\delta D$ and $\delta^{18}O$ exhibiting higher sensitivity to seismic activity.

We have added the following content about method application prospects to the last part of the 4.4. Limitations and prospects section:

Owing to significant differences in hydrogeological settings, tectonic activity, and the current limitations in quantitatively modeling geothermal water circulation under specific geological conditions, a universal set of model parameters applicable across all hot springs within even the same tectonic region cannot be established. This highlights the necessity of anomaly detection model, which involves optimizing parameters specifically for individual hot springs based on their unique pre-seismic responses in different hydrochemical components. The model aims to leverage the inherent differences among these hydrochemical components, integrating them to enhance forecasting efficacy. Crucially, this methodological framework is transferable. For application in other tectonic regions, the model can be adapted by similarly optimizing the parameter combinations for the target hot spring(s) based on their specific hydrochemical components. This addresses the challenge posed by varying tectonic and hydrogeological conditions leading to divergent hydrochemical behaviors. By enabling the application of the model to hot spring monitoring in specific regions through this targeted parameter optimization, the model provides an attempt at a method to advancing earthquake forecasting.

**References:**

Feng, X., Zhong, J., Yan, R., Zhou, Z., Tian, L., Zhao, J., and Yuan, Z.: Groundwater Radon Precursor Anomalies Identification by EMD-LSTM Model, water, 14(1),

https://doi.org/10.3390/w14010069, 2022.

Skelton, A., Sturkell, E., Mörth, C.-M., Stockmann, G., Jónsson, S., Stefansson, A., Liljedahl-Claesson, L., Wästeby, N., Andrén, M., Tollefsen, E., Gunnarsson Robin, J., Keller, N., Geirsson, H., Hjartarson, H., and Kockum, I.: Towards a method for forecasting earthquakes in Iceland using changes in groundwater chemistry. Communications Earth & Environment, 5(1), https://doi.org/10.1038/s43247-024-01852-3, 2024.

6. *#The discussion attributes hydrochemical anomalies to stress-induced fluid mixing and rock dissolution. However, it fails to propose specific geochemical pathways, such as mineral dissolution kinetics or isotopic fractionation, to explain the heightened sensitivity of certain components (e.g., δD, δ¹⁸O). This lack of mechanistic insight limits the study's contribution to understanding earthquake preparation processes and weakens its theoretical foundation.*

   **Response:** Thank you for your comment. Regarding the general mechanisms behind hydrochemical anomalies, it is recognized that pre-earthquake stress accumulation leads to fresh mineral surface exposure during micro-fracturing, which enhances water–rock interaction and can increase ion concentrations as well as $\delta^{18}O$ values (a process often termed $\delta^{18}O$ shift). For example, the rise in $Na^+$ concentrations before earthquake may result from a switchover to nonstoichiometric dissolution of analcime at fresh rock surface with preferential release of $Na^+$ into groundwater (Andrén et al., 2016). Similarly, the increase in $\delta^{18}O$ before the *M* 6.6 Tottori earthquake in southwestern Japan has been attributed to enhanced water–rock interaction due to rock strain during the earthquake preparation process (Onda et al., 2018). Furthermore, fluid mixing following aquifer breaching is also a widely accepted mechanism for pre-earthquake hydrochemical anomalies. The mixing of fluids with significantly different isotopic and hydrochemical compositions can cause variations in $\delta D$, $\delta^{18}O$, and ion concentrations. Previous studies in Iceland have reported pre-earthquake increases in

Na$^+$ concentration, δD, and δ$^{18}$O due to mixing with different groundwater (Skelton et al., 2014; Skelton et al., 2024). A similar phenomenon includes elevated electrical conductivity (EC) and ion concentrations before the $M$w5.0 Mudanya earthquake in western Turkey (Yakupoglu et al., 2025). In contrast, post-seismic periods often involve mixing with shallow water, which may lead to decreases in ion concentrations and isotopic values, as observed after the $M$w7.0 Kumamoto earthquake in Japan (Hosono et al., 2020) and the $M$5.0 Tonghai earthquake in China (Shi et al., 2020). These cases represent detailed, event-specific mechanistic studies. However, since the present study focuses on anomaly detection across multiple earthquakes with diverse hydrochemical changes, it remains difficult to identify a universal mechanism that explains all anomalies. Therefore, based on the dominant patterns reported in previous research, our model was designed to detect high-value anomalies, which better reflect typical pre-earthquake observations.

The higher sensitivity of δD and δ$^{18}$O may be attributed to the fact that stable water isotopes are more conservative than hydrochemical ions. After fluid mixing, the isotopes are not easily altered by short-term chemical processes such as dissolution or precipitation, and instead follow a simple mixing process. In contrast, the ion concentrations can be easily influenced by multiple concurrent processes, such as precipitation/dissolution, cation exchange, adsorption/desorption, and redox reactions, which cause the thermal water tend to ion re-equilibrate, making their responses to earthquakes more complex and less sensitive.

**References:**

Andrén, M., Stockmann, G., Skelton, A., Sturkell, E., Mörth, C. M., Guðrúnardóttir, H. R., Keller, N. S., Odling, N., Dahrén, B., Broman, C., Balic-Zunic, T., Hjartarson, H., Siegmund, H., Freund, F., & Kockum, I.: Coupling between mineral reactions, chemical changes in groundwater, a nd earthquakes in Iceland. Journal of Geophysical Research: Solid Earth, 121(4), 2315-2337. https://doi.org/10.1002/2015jb012614, 2016.

Hosono, T., Yamada, C., Manga, M., Wang, C. Y., and Tanimizu, M. Stable isotopes show that earthquakes enhance permeability and release water from mountains, Nat Commun, 11(1), 2776, https://doi.org/10.1038/s41467-020-16604-y, 2020.

Onda, S., Sano, Y., Takahata, N., Kagoshima, T., Miyajima, T., Shibata, T., Pinti, D. L., Lan, T., Kim, N. K., and Kusakabe, M.: Groundwater oxygen isotope anomaly before the M6.6 Tottori earthquake in Southwest Japan. Scientific Reports, 8(1), 4800. https://doi.org/10.1038/s41598-018-23303-8, 2018.

Shi, Z., Zhang, H., and Wang, G.: Groundwater trace elements change induced by M5.0 earthquake in Yunnan, Journal of Hydrology, 581, https://doi.org/10.1016/j.jhydrol.2019.124424, 2020.

Skelton, A., Andrén, M., Kristmannsdóttir, H., Stockmann, G., Mörth, C. M., Sveinbjörnsdóttir, Á., Jónsson, S., Sturkell, E., Guðrúnardóttir, H. R., Hjartarson, H., Siegmund, H., and Kockum, I.: Changes in groundwater chemistry before two consecutive earthquakes in Iceland, Nature Geoscience, 7(10), 752-756, https://doi.org/10.1038/ngeo2250, 2014.

Skelton, A., Sturkell, E., Mörth, C.-M., Stockmann, G., Jónsson, S., Stefansson, A., Liljedahl-Claesson, L., Wästeby, N., Andrén, M., Tollefsen, E., Gunnarsson Robin, J., Keller, N., Geirsson, H., Hjartarson, H., and Kockum, I.: Towards a method for forecasting earthquakes in Iceland using changes in groundwater chemistry. Communications Earth & Environment, 5(1), https://doi.org/10.1038/s43247-024-01852-3, 2024.

Yakupoğlu, N., Sabuncu, A., Erbil, C., Kırkan E., Çetin H., and İnan S.: Pre-earthquake hydrogeochemical anomalies in spring waters: two distinctive cases from western Türkiye, Journal of Hydrology, 662, https://doi.org/10.1016/j.jhydrol.2025.133920, 2025.

7.  *#The discussion asserts that the model outperforms single-component methods and compares favorably to Zhu et al. (2024). However, the reported false alarm rate*

*(FAR, ~0.28–0.33) is relatively high for practical forecasting applications. The absence of statistical tests to confirm the model's superiority, combined with an overemphasis on TS and POD without addressing the operational impact of false alarms, overstates the model's practical utility.*

**Response:** We thank the reviewer for this comment. Our statement that the model "outperforms single-component methods" refers exclusively to an internal comparison within our own model's framework. As shown in Figures 7, 8, and 9, the multi-component joint warning result demonstrably achieves a higher TS and POD values than the result from any single-component used in isolation. This improvement represents the central thesis of our method: integrating multiple hydrochemical components mitigates limitations and environmental noise inherent in individual components, thereby highlighting the unique advantage of multi-component synergy in constructing a more robust and accurate anomaly detection model.

Our intention was not to claim superiority over Zhu et al. 2024, but to use their recent and comprehensive study as a benchmark to demonstrate that our model's performance (POD: 0.83–0.95, TS: 0.59–0.70) is comparable to they reported best-performing method (LOF, POD of ~0.7, R-score of ~0.6, which is analogous to a TS of ~0.6).

We fully agree with the reviewer that a FAR of 0.28–0.33 is a significant consideration for operational forecasting, and we did not intend to understate this point. This high FAR reflects a challenging trade-off in anomaly detection, wherein maximizing the POD often incurs a higher FAR. The primary contribution of our model is its enhanced sensitivity to potential anomalies through a multi-component approach, as demonstrated by the high POD and TS. The TS serves as an integrated metric that balances both POD and FAR, any increase in FAR would correspondingly lower the TS. We acknowledge that high FAR remains a common challenge in the field, which often prioritizes sensitivity to avoid missing potential anomalies. Therefore, our model is designed not as a standalone forecasting tool, but as an assistive component aimed at

identifying hydrochemical anomalies. For practical applications, we recommend integrating its outputs with other geophysical, geodetic, and geological data to form a more robust and reliable seismic risk assessment.

In the revision, we have revised the manuscript on line 463 to address lack of clarity and to avoid any overstatement of utility:

At QJ, the model provides 21 effective warnings for 22 earthquake events (POD = 0.95), with 8 false alarms (FAR = 0.28) and a TS of 0.70. At WN, the model generates 10 accurate warnings for 12 events (POD = 0.83), 5 false alarms (FAR = 0.33), and a TS of 0.59. It is noteworthy that while the obtained FAR remains relatively high, which is a common challenge in earthquake anomaly detection that prioritize detection sensitivity, the TS provides a comprehensive metric that balances both POD and FAR. Compared with the internal single-component anomaly detection results from our model, the multi-component joint warning results exhibit higher TS values (Figures 7, 8, 9). This observation demonstrates that multicomponent collaboration mitigates the effects of geochemical behavior differences among components, reduces environmental interference on individual ions/ion pairs, and consequently enhances the accuracy of the anomaly detection model. Zhu et al. (2024) comprehensively evaluated the anomaly detection performance of several machine learning algorithms using 2.5 years of hydrochemical data from the southeast coast of China. The best-performing local outlier factor algorithm achieved an R-score of about 0.6, POD of about 0.7, and FAR of about 0.15. The improved anomaly detection model demonstrates comparable performance, which confirms its effectiveness. These results indicate the practical value of the multi-component model for anomaly identification, though its practical application would benefit from integration with other geophysical, geodetic, and geological data to further reduce the false alarm burden.

8. *#Line 239: The manuscript states that a 15-day moving average is applied to 3-day resolution hydrochemical data, implying only five measurements per 15-day*

*period. The rationale for averaging over 15 days is not explained, which raises questions about the appropriateness of this window size for capturing tectonic signals while filtering noise. A justification or sensitivity analysis for this choice is needed to ensure methodological rigor.*

**Response:** Thank you for your insightful comment regarding the choice of the 15-day moving average. The use of a moving average over this timescale (e.g., 14-day) is a well-established methodology in fluid geochemical analysis, as evidenced by its application in numerous previous studies (Piersanti et al., 2016; Fu et al., 2017; Zhao et al., 2021). We use a 15-day window, which is more applicable to 3-day resolution data. Furthermore, the model is applied to enhance the capability for short-term and imminent earthquake forecasting (within a 45-day window). So, the moving average window time is set to be shorter than the earthquake response time threshold (45 days).

To ensure methodological rigor, we have added a justification for the 15-day moving average at line 235 of the revised manuscript:

Unlike temperature and pressure, rainfall causes pulsed disturbances, typically manifesting as intermittent spikes followed by extended zero-value intervals in sampling data. Previous studies typically employ a 14-day moving average to filter out such interference, a method that has been established as effective in geochemical analysis (Piersanti et al., 2016; Fu et al., 2017; Zhao et al., 2021). To suppress high-frequency noise from short-term environmental disturbances such as rainfall while preserving mid- to low-frequency tectonic signals, a 15-day backward moving average is applied to process the 3-day resolution hydrochemistry data.

**References:**

Fu, C., Yang, T., Tsai, M., Lee, L., Liu, T., Walia, V., Chen, C., Chang, W., Kumar, A., and Lai, T.: Exploring the relationship between soil degassing and seismic activity by continuous radon monitoring in the Longitudinal Valley of eastern Taiwan, Chemical Geology, 469, 163-175, https://doi.org/10.1016/j.chemgeo.2016.12.042,

2017.

Piersanti, A., Cannelli, V., and Galli, G.: The Pollino 2012 seismic sequence: clues from continuous radon monitoring, Solid Earth, 7(5), 1303-1316, https://doi.org/10.5194/se-7-1303-2016, 2016.

Zhao, Y., Liu, Z., Li, Y., Hu, L., Chen, Z., Sun, F., and Lu, C.: A case study of 10 years groundwater radon monitoring along the eastern margin of the Tibetan Plateau and in its adjacent regions: Implications for earthquake surveillance, Applied Geochemistry, 131, https://doi.org/10.1016/j.apgeochem.2021.105014, 2021.

9. *Figure 4: For the QJ station, the manuscript reports 22 $M \geq 4$ earthquakes, yet Figure 4a displays only 11 earthquakes. This discrepancy is unacceptable and suggests incomplete data visualization. Similar inconsistencies appear in other subfigures, undermining the reliability of the visual representation of results and necessitating a comprehensive review of figure accuracy.*

**Response:** Thank you for raising this important point. In the original Figure 4a, the 11 $M \geq 4$ earthquakes shown were not meant to represent all recorded earthquake events, but specifically those that occurred after the detected Bayesian change points and were therefore identified as successfully forecasted by the model. This display intended to maintain visual clarity in illustrating the temporal relationship between change points and subsequent earthquakes. The evaluation metrics provided alongside the figure already offer a comprehensive assessment of forecasting performance, including both successful predictions and misses, thereby fully reflecting the effectiveness of the Bayesian change point analysis.

However, we fully acknowledge that displaying all earthquakes improves transparency and allows readers to visually correlate all seismic activity with the identified change-points. In response to your feedback, we have revised the figure to include all $M \geq 4$ earthquakes. Successfully detected earthquake events are now indicated in black, while those not detected by the model are shown in gray. Similar

revisions have been consistently applied to all other relevant subfigures.

The revised figure is as follows:

[Figure]

Figure 4. Anomaly detection results from the Bayesian change point (BCP) analysis applied to hydrochemical component time series. The black solid line represents the component concentration after 15-day moving averaging. The green dashed line indicates the forecasting model of the BCP algorithm. The red solid line shows the posterior probability of change points. Yellow stars mark earthquake events. Black and gray vertical bars show detected and missed earthquakes respectively. The false alarm rate (FAR), probability of detection (POD), and threat score (TS) are evaluation

metrics.

**Mirror comments:**

1.  *#Line 25: The phrase "tailored model parameters for specific hydrochemical components" is imprecise. It should specify that parameters are optimized for individual components (e.g., $Na^+$, $\delta^{18}O$) based on their distinct geochemical responses to seismic stress, as elaborated later (lines 428–431).*

    **Response:** Thank you for this insightful comment. We have revised the phrasing on Line 25:

    Parameters are optimized for individual components based on their distinct geochemical responses to seismic stress, thereby significantly enhancing the model's performance and adaptability.

2.  *#Line 42: The term "physicochemical properties" is overly general. To align with the study's focus on hydrochemical components, specify the properties primarily affected by crustal stress changes, such as ion concentrations and isotopic ratios.*

    **Response:** Thank you for the insightful comment. We agree that "physicochemical properties" was too general. To thoroughly assess the impact of crustal stress changes on fluid properties and to highlight the response characteristics of hydrochemical components (such as ion concentrations and stable isotope ratios), we supplemented case evidence on hydrochemistry. Additionally, the originally cited literature on water-level variations has been replaced with references pertaining to changes in stable isotopes of water.

    These responses often result in significant changes in the physical and chemical properties of the fluids, such as their ion concentrations and isotopic ratios (Gori and Barberio, 2022; Tian et al., 2023; Skelton et al., 2024).

    **References:**

Skelton, A., Sturkell, E., Mörth, C.-M., Stockmann, G., Jónsson, S., Stefansson, A., Liljedahl-Claesson, L., Wästeby, N., Andrén, M., Tollefsen, E., Gunnarsson Robin, J., Keller, N., Geirsson, H., Hjartarson, H., and Kockum, I.: Towards a method for forecasting earthquakes in Iceland using changes in groundwater chemistry. Communications Earth & Environment, 5(1), https://doi.org/10.1038/s43247-024-01852-3, 2024.

3.  *#Line 78: The statement "thermal springs tend to exhibit high stability" may mislead readers, as stability is context-specific. Clarify that this refers to their low susceptibility to short-term environmental fluctuations (e.g., temperature) compared to other fluid systems.*

**Response:** We appreciate you highlighting the need for greater precision. We have revised the manuscript to explicitly state that the stability of thermal springs is discussed in comparison to other fluid systems, particularly near-surface groundwater, and to clarify that this stability refers to resilience against short-term environmental fluctuations. The revised content on Line 78:

The hydrochemical components (e.g., $Na^+$, $Cl^-$, $SO_4^{2-}$) of thermal springs tend to exhibit greater stability against short-term environmental fluctuations (e.g., temperature, short-term rainfall) compared to near-surface cold water systems, alongside deep circulation depth, rapid upward migration and limited susceptibility to anthropogenic influence. These characteristics help minimise non-seismic noise and allow for a more accurate reflection of hydrogeological changes during earthquake preparation (Martinelli, 2020; Tian et al., 2024).

4.  *#Line 96: The reference to Piersanti et al. (2016) is introduced abruptly without clarifying its relevance to hydrochemical data. Briefly note that the algorithm, originally developed for radon time series, was adapted for multicomponent hydrochemical analysis to enhance reader comprehension.*

**Response:** Thank you for this helpful suggestion. We have revised the manuscript to clarify the relevance of the algorithm by Piersanti et al. (2016). The sentence on Line 96 has been modified to better explain its original purpose and our adaptation:

The anomaly detection algorithm originally developed by Piersanti et al. (2016) for radon time series was adapted for real-time multicomponent hydrochemical analysis in thermal springs within the study area.

5. *#Line 155: The description of the water quality analyzer (HQ40D, HACH, USA) and its measurement accuracies (0.1°C, 0.01 pH, 1 μS/cm) is tangential to the study's primary focus. Omit or briefly summarize this detail to maintain emphasis on the hydrochemical data.*

**Response:** Thank you for this valuable suggestion. As per your comment, we have revised the manuscript (lines 152-156) accordingly.

Water temperature, pH, electrical conductivity (EC) and hydrochemical components for the thermal springs were measured every three days.

6. *#Line 163: The list of analyzed ions is overly comprehensive. Specify only the ions used in the study to maintain focus and avoid extraneous detail.*

**Response:** Thank you for your thoughtful comment. The comprehensive ion list was included to ensure full data transparency and to perform essential ion-balance checks, which are critical for validating measurement quality. The reasons for selecting the specific ions used in our analysis, as opposed to this full dataset, are explicitly detailed in the manuscript (Line 217). We believe this approach upholds methodological rigor while maintaining a focused narrative.

7. *#Line 172: The ion balance error equation is presented but not referenced or applied elsewhere in the study, rendering it disconnected from the analysis. Clarify*

*its use or remove it to avoid confusion.*

**Response:** Thank you. We agree that the purpose of the ion balance error equation should be made clear in the text. To address this, we have revised the manuscript to explicitly state that the calculation was used for quality control purposes, ensuring the analytical reliability of the data prior to any further analysis. Furthermore, reference and parameter clarifications are provided on lines 170:

To ensure data accuracy, cation–anion balance error tests were performed for each sample as a quality control measure, with all ionic deviations kept within ± 5%, and data fulfilling this criterion were included in the subsequent analysis. The ion balance error (Appelo and Postma, 2004) is calculated as below:

$$ib(\%) = \frac{\sum cations - \sum anions}{\sum cations + \sum anions} \times 100 \tag{1}$$

where $\sum cations$ represents the sum of cation concentrations (in milliequivalents per liter, meq/L), and $\sum anions$ represents the sum of anion concentrations.

**References:**

Appelo, C.A.J., and Postma, D.: Geochemistry, Groundwater and Pollution (2nd ed.), A.A. Balkema Publishers, Leiden, 17pp, ISBN04 1536 428 0, 2004.

8.  *#Line 175: The claim that all earthquakes with M ≥ 4 are "destructive" is inaccurate, as destructiveness depends on depth, location, and infrastructure. Revise to reflect that M ≥ 4 earthquakes are the study's focus without implying universal destructiveness.*

**Response:** We fully agree with your suggestion. To ensure more precise expression, we have removed "destructive" and the revised version is as follows:

The anomaly detection model developed in this study focused on forecasting (identifying anomalous signals preceding) earthquakes with magnitudes (M) ≥ 4.

9.  *#Line 188: The text states that QJ was within the preparation zones of 22 M ≥ 4*

*earthquakes, but Table S1 lists 24 events, creating a discrepancy. Clarify the correct number in the text to ensure consistency.*

**Response:** We sincerely appreciate the reviewer's meticulous attention to data consistency. Upon thorough verification:

**Table S1 correctly lists 22 events for QJ**: While Table S1 includes 24 total M ≥ 4 earthquakes (sorted chronologically), only 22 of these events have preparation zones encompassing the QJ site. We apologize for any ambiguity caused by the chronological sorting of Table S1.

**The manuscript text (22 events) is accurate**: The statement "QJ was within the preparation zones of 22 M ≥ 4 earthquakes" remains valid and consistent with Table S1 data.

**Table S1.** Catalog of earthquakes meeting the selection criteria during the thermal spring monitoring period.

| Date | Lon. | Lat. | Depth (km) | M | Δ (km) | | Response sites |
|---|---|---|---|---|---|---|---|
| | | | | | QJ | WN | |
| 2019/06/24 | 101.64 | 24.93 | 10 | 5.2 | 161 | − | QJ |
| 2019/08/31 | 101.95 | 23.34 | 13 | 4.3 | 109 | − | QJ |
| 2019/11/01 | 102.79 | 24.39 | 13 | 4.0 | 51 | − | QJ |
| 2020/01/15 | 103.12 | 25.55 | 8 | 4.8 | 182 | − | QJ |
| 2020/01/23 | 101.86 | 23.37 | 15 | 4.0 | 115 | − | QJ |
| 2020/04/11 | 101.89 | 23.67 | 12 | 4.1 | 97 | − | QJ |
| 2020/06/16 | 102.72 | 22.64 | 9 | 5.2 | 144 | − | QJ |
| 2020/07/12 | 102.52 | 22.86 | 11 | 4.8 | 123 | − | QJ |
| 2021/05/21 | 99.88 | 25.70 | 10 | 6.7 | 354 | − | QJ |
| 2021/06/10 | 101.92 | 24.35 | 8 | 5.6 | 100 | − | QJ |
| 2021/06/16 | 101.90 | 24.34 | 8 | 4.8 | 101 | − | QJ |
| 2021/06/28 | 101.89 | 24.31 | 8 | 4.9 | 101 | − | QJ |
| 2021/11/16 | 101.68 | 22.32 | 10 | 5.2 | 213 | 135 | QJ/WN |
| 2021/12/24 | 101.68 | 22.34 | 10 | 6.6 | 211 | 133 | QJ/WN |
| 2022/03/05 | 101.63 | 22.37 | 8 | 4.8 | 191 | 131 | QJ/WN |
| 2022/07/22 | 99.90 | 21.10 | 10 | 6.2 | 434 | 338 | QJ/WN |
| 2022/09/05 | 102.09 | 29.59 | 16 | 6.8 | 632 | 676 | QJ/WN |
| 2022/11/19 | 102.29 | 23.40 | 8 | 5.4 | 79 | 42 | QJ/WN |

| | | | | | | | |
|---|---|---|---|---|---|---|---|
| 2022/11/21 | 102.27 | 23.43 | 14 | 4.1 | – | 39 | WN |
| 2023/03/03 | 102.60 | 22.55 | 10 | 4.6 | 156 | 129 | QJ/WN |
| 2023/03/23 | 100.69 | 22.62 | 10 | 4.7 | – | 159 | WN |
| 2023/05/31 | 102.65 | 24.20 | 16 | 4.0 | 33 | 108 | QJ/WN |
| 2023/08/13 | 101.86 | 24.32 | 10 | 4.8 | 104 | 89 | QJ/WN |
| 2023/11/17 | 99.35 | 21.20 | 10 | 6.2 | 467 | 368 | QJ/WN |

" – " means no data.

10. *#Line 214: Figure 2 is referenced without specifying its content (e.g., time series of which components). Clarify that it illustrates hydrochemical component time series (e.g., Na⁺, Ca²⁺, Cl⁻) alongside rainfall and earthquake events for QJ spring to guide readers.*

    **Response:** We thank the reviewer for this helpful suggestion. The figure captions have been revised to explicitly state the content of Figure 2 and Figure S1. The captions now read:

[Figure]

Figure 2. Time series of hydrochemical components (Na⁺, K⁺, Ca²⁺, Cl⁻, SO₄²⁻, HCO₃⁻, F⁻, δD, and δ¹⁸O), alongside corresponding rainfall and earthquake events for Qujiang spring.

Figure S1. Time series of hydrochemical components (Na⁺, K⁺, Ca²⁺, Cl⁻, SO₄²⁻, HCO₃⁻, F⁻, δD, and δ¹⁸O), alongside corresponding rainfall and earthquake events for Wana spring.

11. *Line 299: The evaluation metrics (FAR, POD, TS) are introduced in Figure 4's caption but not defined until later (lines 370–380). Define them first before using them to avoid confusion for readers encountering the metrics early.*

**Response:** We thank the reviewer for highlighting this logical flow issue. The results of BCP analysis (Lines 295-314) have been relocated from the original location

to the Results and discussion section. The metrics are formally defined immediately before their first appearance in Figure 4 (now in Results). The caption retains the abbreviated names after their full definitions in text, maintaining visual clarity.

*12. Figures 7 and 8 effectively present anomaly detection results, but their captions and annotations lack sufficient detail. The figures omit scales or legends for posterior probabilities. Revise captions to include legends to enhance accessibility and enable independent verification of results.*

**Response:** Thank you for your valuable feedback regarding the clarity of Figures 7 and 8. We would like to clarify that Figures 7 and 8 specifically present the primary results of the anomaly detection model, while the results from the Bayesian method, which serves a supplementary analysis, are separately illustrated in Figure 4. Our initial intention was to avoid overcrowding the figures and to maintain a clear focus on the respective analyses. Integrating all results into a single figure did, after our try, result in visual clutter. To improve coherence, we have repositioned Figure 4 (Bayesian results) immediately before Figure 7 in the revised manuscript. This arrangement ensures that both sets of results are logically organized and enable independent verification.

*13. Line 520: The discussion references "multiple mechanisms" for anomalies (e.g., Thomas, 1988) without specifying examples, such as fracture dilation or fluid mixing. Briefly list one or two mechanisms to clarify the context.*

**Response:** Thank you for this insightful comment. We have revised the manuscript to explicitly include these mechanisms for clarity on Line 518:

This observation underscores the complex dynamic mechanisms and regional structural differences involved in the earthquake preparation process. The geochemical anomalies often arise from the combined effects of multiple mechanisms, such as fluid mixing following aquifer breaching or fresh mineral surface exposure during micro-fracturing, resulting in increased hydrochemical component concentrations (Thomas,

1988).

14. *Line 574: The phrasing "tailored model parameters… account for their differences" is awkward and lacks clarity. Streamline for precision and readability.*

**Response:** We thank the reviewer for this valuable feedback. We agree and have revised the sentence on Line 574 to improve its clarity and flow.

Parameters are optimized for individual components based on their distinct geochemical responses to seismic stress, significantly enhance the model's performance and adaptability.